# On Defining Neural Averaging

**Su Hyeong Lee**[*]
Department of Statistics
University of Chicago

**Richard Ngo**
Independent

## Abstract

What does it even mean to average neural networks? We investigate the problem of synthesizing a single neural network from a collection of pretrained models, each trained on disjoint data shards, using only their final weights and no access to training data. In forming a definition of neural averaging, we take insight from model soup, which appears to aggregate multiple models into a singular model while enhancing generalization performance. In this work, we reinterpret model souping as a special case of a broader framework: Amortized Model Ensembling (AME) for neural averaging, a data-free meta-optimization approach that treats model differences as pseudogradients to guide neural weight updates. We show that this perspective not only recovers model soup but enables more expressive and adaptive ensembling strategies. Empirically, AME produces averaged neural solutions that outperform both individual experts and model soup baselines, especially in out-of-distribution settings. Our results suggest a principled and generalizable notion of data-free model weight aggregation and defines, in one sense, how to perform neural averaging.

## 1 Introduction

Consider the following problem. Suppose we have $N$ neural networks that are domain experts with differing capabilities, perhaps due to being trained on different shards of a larger data stream. Therefore, each expert is knowledgeable about its own shard, but is largely oblivious or out-of-domain on others. Now, remove all access to the data. We are only given the model weights and nothing else. In this setup, can we synthesize a single model that preserves the knowledge embedded in each expert, using only these final weights and without performing any additional data-driven training?

At a high level, our goal is to construct a form of "average" over these domain experts–a model that aggregates their capabilities and ideally retains their individual performance. But what does it even mean to perform neural averaging? What properties should such an average of neural nets satisfy?

In this work, we propose a framework for answering these questions. We begin by identifying a characteristic for neural network averaging. Neural nets are often highly specialized: they memorize the data they are trained on and tend to generalize less well outside of it [1, 2]. If we succeed in constructing an averaged model, then, even without access to any training data, we expect it to retain the memory of each expert. That is, such a model should perform better on each expert's original training distribution than it would on a random validation set, despite never having seen any of that data itself.

Is such a notion of neural averaging even realizable? A naive approach might be to create a rudimentary mixture model of neural nets: at inference time, sampling from a multinomial distribution over the $N$ experts and using the selected model to make predictions. In expectation, this creates a kind of functional ensemble. However, this strategy is deeply unsatisfying. It requires constantly maintaining access to all $N$ models, which is inefficient, and does nothing to address the fact that

---

[*]Correspondence to: `sulee@uchicago.edu`

each expert is blind to most of the overall data distribution. Deploying any single expert is thus inherently brittle.

A more promising direction comes from the model soup literature [3, 4]. These works show that averaging the weights of multiple independently trained models, e.g., trained with different data orderings or hyperparameters, can yield a new model with stronger generalization. Not only does this suggest that weight-space averaging can be meaningful, but also that it can outperform the individual models being averaged. This empirical success motivates our central question: can we extend model soup to a useful notion of neural averaging?

Our approach is guided by the following intuition. During training, each minibatch gradient update pushes a model through a high-dimensional, non-convex optimization landscape, nudging its weights to encode the statistical features of the data [5, 6]. After enough iterations, the model converges to a local minimum that reflects the distribution it was trained on. We hypothesize that these converged weights implicitly contain enough directional signal that, by computing distance vectors in weight space, we can construct pseudogradients that continue to point toward the expert's learned minima. If this is true, then we may be able to define a meta-optimization algorithm, operating directly on the expert weights, that mimics the effect of data-driven training. Crucially, this approach would require no actual access to data, relying only on the geometry of the converged models. Moreover, if we employ adaptive optimizers known to accelerate convergence in non-convex settings [7, 8, 9], we may be able to reach high-quality solutions more efficiently [10, 11].

In this paper, we introduce a novel ensembling algorithm that aggregates multiple pretrained neural networks into a single model, without using any data or retraining. Our method constructs updates in weight space by treating model differences as pseudogradients, enabling optimization toward a fused model that integrates the knowledge of each expert. We show that the state-of-the-art model soup strategy [3, 4] emerges as a special case of our framework, corresponding to a single-step gradient descent with a fixed step size. By reframing ensembling as data-free meta-optimization, we offer a new perspective on what it means to "average" in neural network space, that is, to perform neural averaging.

## 2   Related Literature

### 2.1   Ensembling Methods in Machine Learning

Traditional ensembling techniques seek to combine multiple models to enhance overall performance, leveraging the strengths of each individual model. Bootstrap Aggregating [12] trains multiple instances of the same model on different subsets of the training data created through random sampling with replacement. This approach can help reduce variance and mitigate overfitting [13, 14]. Boosting [15, 16], on the other hand, builds models sequentially, where each new model attempts to correct the errors of its predecessors. Voting [17] combines the predictions of multiple models by taking a majority vote for classification tasks or averaging for regression tasks. Although these methods are effective in improving model robustness and predictive accuracy, bagging and boosting are expensive to implement for DNN architectures, and voting requires the maintenance of all selected workers in parallel, resulting in an $n$-fold increase in run-time compute for $n$ ensembled learners during deployment [3]. In contrast, our approach consolidates all $n$ sub-optimal learners into a single, powerful learner.

### 2.2   Model Ensembling in Deep Learning & Model Soup

The idea of combining multiple neural network weights, either through weighted linear averaging or exponential moving averaging, is a well-studied approach [18, 19, 20, 21, 22, 23, 24]. For instance, Snapshot Ensembling [25] captures multiple local minima in the optimization landscape over a single training run, by encouraging transitory local convergence via a periodic warm-up-ramp-down learning rate schedule. The state dictionary at each minima is saved, and the last $m$ snapshots are linearly averaged to form a robust predictive model, improving generalization without additional training cost. Additional works have shown that when averaging models that share a portion of the optimization trajectory, including fine-tuning with pre-trained initialization, model accuracies typically do not degrade significantly when averages are taken [26, 27].

A recent development is model soup [3, 22], which showed that in pre-trained transfer learning settings, linear averaging of model parameters (called *soup ingredients*) can make substantial gains on out-of-distribution performance and zero-shot capabilities on downstream tasks. In contrast with previous approaches, souping merges models fine-tuned with hyperparameter configuration diversity (e.g., momentum parameters and learning rates). A recent follow-up work [23] shows that linear interpolation between the weights of fine-tuned models uncovers domains of comparable performance within the simplex whose vertices are the soup ingredients. Despite these empirical observations, it remains unclear why such a linear aggregation of model weights ensembles effectively. In this paper, we show that this procedure can be realized as a very specific sub-category of amortized model ensembling with adaptivity disabled. This entirely optimization-driven approach taken in our work differs significantly from previous interpretations of the superior performance attained via DNN model interpolation, where under strict assumptions, averaging weights discovered over SGD [28] trajectories was viewed as approximately sampling from a Gaussian and computing the maximum likelihood estimator [18, 29].

## 3 Reformulation of Model Aggregation as Gradient Descent

### 3.1 Understanding Weight Averaging as a Single Optimization Step

In this section, we show that the linear model averaging process is equivalent to optimizing over an online quadratic loss objective, with the optimum located precisely at each random variable $\xi$ drawn from the population distribution $\mathcal{D}$ of soup ingredients. Therefore, if $\mathcal{D}$ is a single distribution over well-trained models with limited variance, such as over a loss landscape which is empirically well-approximated by a quadratic convex objective, we may intuit that the souped model should display strong performance. In particular, while the optimization of DNNs is a non-convex problem [5], Goodfellow et al [30] observes that loss surfaces can sometimes be approximately convex in practice, over a single trajectory when optimized with SGD. However, if $\mathcal{D}$ is a mixture model comprising multiple distributions associated with well-trained learners situated in disparate regions of the non-convex loss landscape, the simplistic quadratic objective function will exhibit significant fluctuations depending on the sampled distribution. This increased variation implies that non-adaptive gradient descent on a basic quadratic loss will struggle to stabilize the souped output, leading to sub-optimal performance. We now motivate the AME framework by formalizing model soup.

Consider the simple stochastic loss function,

$$f(x) = \frac{1}{2} \left( ||x - \xi||^2 - ||\xi||^2 \right), \tag{1}$$

where $x \in \mathbb{R}^d$ is the high-dimensional space of model parameters and $\xi \sim \mathcal{D}$ for $\mathcal{D}$ the distribution over trained model parameters. After drawing $N$ models $\xi_1, \ldots, \xi_N$ to materialize $N$ (online) realizations of the stochastic objective, we perform gradient descent after initializing[2] at an *arbitrary* $x_{pivot}$ with harmonic learning rate decay $\eta_i = 1/i$, at step $i$. Letting $x_i = \xi_i$ for notational clarity, the first step of gradient descent gives

$$w_1 = x_{pivot} - \frac{1}{1}(x_{pivot} - x_1),$$

which leads to the second step

$$w_2 = x_1 - \frac{1}{2}(x_1 - x_2) = \frac{1}{2}(x_1 + x_2).$$

Noting the identity

$$\frac{1}{n} \left( \sum_i^n x_i \right) - \frac{1}{n+1} \left( \frac{1}{n} \left( \sum_i^n x_i \right) - x_{n+1} \right) = \frac{1}{n+1} \left( \sum_i^{n+1} x_i \right),$$

we obtain after $N$ steps the souped output

$$x_{soup} = w_N = \frac{1}{N} \left( \sum_{i=1}^N x_i \right). \tag{2}$$

---

[2]Alternatively, we may initialize at ingredient $x_1$ and choose the learning rate schedule $\eta_i = 1/(i + 1)$.

Therefore in this paper, we evaluate the utility of regularizing neural nets over the simple objective (1), whose optimum is located at each $\xi \sim \mathcal{D}$. As there exist many distributions such that the empirical average deviates from the maximum likelihood estimator, we seek to utilize stochastic adaptive optimization that effectively supports a diverse range of $\mathcal{D}$ to locate a suitable ensembled model $\xi_{ens} \neq x_{soup}$.

Therefore, we reinterpret model souping as regularization over an online convex objective which is non-adaptively optimized for one epoch in expectation at $\mathbb{E}[\xi]$, the first moment of $\mathcal{D}$. This paves way for the deployment of alternative learning descent algorithms (e.g., Adam [7], Adagrad [8]) for neural averaging, which may provide qualitatively variant results. Intuitively, we posit that the ensemble performance is conditional on the distribution $\mathcal{D}$ of trained model weights. For instance, Lee et al [31] (i.e., Theorem 13) show that under the convex stochastic objective (1), if $\mathcal{D}$ follows a heavy-tailed distribution, then the outputs of non-adaptive gradient descent will display infinite variance as well as incur infinite regret in expectation under any learning rate schedule. Therefore, alternative model updates such as clipped-SGD or adaptive optimizers could be used to enhance robustness and accelerate convergence [11, 32, 33]. We include further detailed discussions along with intuition-building synthetic experiments in Appendix F. In Section 3.2, we formalize these notions into a framework which we call amortized model ensembling. We begin by easing out a precise mathematical formulation, which we generalize to subsume the derivation above.

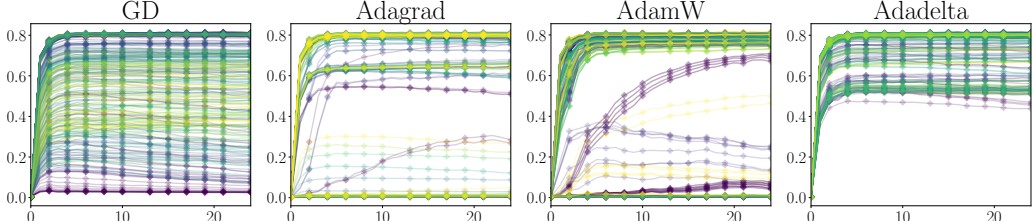

Figure 1: Performance of GD [28], Adagrad [8], AdamW [34], and Adadelta [35] amortized ensembling of ViT-S [36] trained on GLD-23K [37], from left to right, after 1 ensembling epoch. No training nor testing data of any sort are used during the ensemble process (data-free), and only trained model weights are accessed. Plots depict classification accuracy on GLD-23K testing data unseen during training, for a variety of optimizer hyperparameter choices (described in Appendix A). Horizontal axis is the number of model training epochs, different from ensembling epochs, and the color transition indicates the number of ensemble ingredients (models aggregated), ranging from 2 (dark) to 16 (light). This figure shows that on a macroscopic level, each optimizer instantiation induces qualitatively variant ensemble accuracy dynamics, affirming the existence of diverse ensembling strategies uncovered by the AME framework. Details are given in full in Appendix A.2.

## 3.2 Mathematical Formulation of Amortized Model Ensembling

An alternative way of forming a standard model soup is to use a pivot,

$$x_{soup} = \frac{1}{N} \sum_{i \in [N]} x_i = x_{pivot} - \frac{1}{N} \sum_{i \in [N]} (x_{pivot} - x_i) \qquad (3)$$

where $x_1, \ldots, x_N$ are the selected model ingredients. Pivoting adds an additional degree of freedom, which allows the formulation of pivoted pseudogradients. We may equivalently interpret (3) as a sequence of $N$ training steps with initialization $x_{pivot}$ as follows,

$$
\begin{aligned}
w_1 &= x_{pivot} - \frac{1}{N}(x_{pivot} - x_1), \\
w_2 &= w_1 - \frac{1}{N}(x_{pivot} - x_2), \\
&\vdots \\
w_N &= w_{N-1} - \frac{1}{N}(x_{pivot} - x_N).
\end{aligned}
\qquad (4)
$$

Defining $g_i = \zeta_i(x_{pivot} - x_i)/N$ to be the *pivoted pseudogradient* with amplification parameter $\zeta_i$, we formalize this training process as backpropagating with the $g_i$,

$$w_i = w_{i-1} - \eta_i g_i. \qquad \textbf{(GD Ensembling)}$$

Selecting the fixed learning rate $\eta_i = 1$ and gradient amplification parameter $\zeta_i = 1$ results in standard linear averaging. Therefore, we may consider GD backpropagation variants, such as using a learning rate schedule or applying an adaptive optimizer [7, 8] to the pivoted pseudogradients. For instance, we have for unamplified $g_i$,

$$w_i = w_{i-1} - \frac{\eta_i}{\sqrt{\sum_{j=1}^{i} g_j^2} + \epsilon} g_i, \qquad \textbf{(Adagrad Ensembling)}$$

$$w_i = w_{i-1} - \frac{1}{1 - \beta_1^i} \cdot \frac{\eta_i \hat{m}_i}{(\sqrt{1 - \beta_2^i})^{-1}\sqrt{\hat{v}_i} + \epsilon}, \qquad \textbf{(Adam Ensembling)}$$

where for **Adam Ensembling**, the momentum parameters satisfy $\beta_1, \beta_2 \in [0, 1)$ and the moment estimate moving averages are defined as

$$\hat{m}_i = \beta_1 m_{i-1} + (1 - \beta_1)g_i, \quad \hat{v}_i = \beta_2 v_{i-1} + (1 - \beta_2)g_i^2.$$

Note that any linear interpolation of model weights within the simplex whose vertices are the ensemble ingredients follows as a consequence of **GD Ensembling** with an appropriate learning rate schedule.

**Initialization of gradient descent.** In standard SGD of modern DNNs, the weight initialization process typically involves a combination of random generation and informative heuristics to control neuron variance during the early stages of training. This is due to the fact that when a model is trained from scratch, we must often start with a very weak, unsophisticated learner that may wander into undesirable landscapes. Future updates gradually realize stronger learners in an online fashion. By contrast, in ensemble settings, we are typically provided with a wealth of fully realized and suitably trained learners, which necessitates the question of the selection of $x_{pivot}$. Note that $x_{pivot}$ corresponds to the initialization in the gradient descent setting of (4).

**Alternative formulation of Amortized Model Ensembling.** A non-equivalent formulation of AME is to follow the derivation of Section 3.1. Then, the unamplified $i$-th pseudogradient is defined as $(w_i - x_i)/N$, which may be amplified using $\zeta_i$ or an appropriate learning rate schedule. We further detail this setting under the name *adaptive model pivoting* in Section 3.3. This corresponds to choosing the pivot $x_{pivot}$ adaptively at each step when computing pseudogradients, to reflect the latest and most informative learner. In Section 3.3, we develop a concrete amortized ensembling framework using heuristics that guide our algorithm design. Generally, we take smaller learning steps when the model ingredients are weak, and apply larger, more confident updates when the ingredients are strong, determined autonomously by adaptive optimizers which leverage historical gradient statistics.

### 3.3 Amortized Model Ensembling

Given unordered model ingredients $\{x_1, \ldots, x_N\}$, how should we develop the ensemble? In particular, the step-wise construction of gradient descent (4) requires an ordering of the model ingredients to extract the pivoted pseudogradients. How should we initialize the pivot, and what should guide the choice of ordering? For initialization of the pivot, we propose to select an informative model, such as any of the model ingredients $\{x_1, \ldots, x_N\}$ or their standard average (linear soup) as candidates. To order model ingredients, we may construct a list based on any metric such as the validation loss or accuracy on the entire training dataset, from largest to smallest. The ordering can also be task-specific, such as applying a hierarchical total order based on the validation metric evaluated on a particular dataset of interest.

**Adaptive model pivoting.** We now explore concurrently updating the initial pivot $x_{pivot}$ to the best known value, in order to account for suboptimal choices of $x_{pivot}$. A version of this can be done by

---

**Algorithm 1** Amortized Model Ensembling (Unspecified Optimizer)

---

**Require:** Any set of unordered models $\{x_1, \ldots, x_N\} \subset \mathbb{R}^d$, pivot $x_{pivot} \in \mathbb{R}^d$

Learning rate schedule $\eta_i$, pivoted pseudogradient amplification schedule $\zeta_i, \forall i \in [N]$

1: Construct ordered list $x_1, \ldots, x_N$ using metric of choice, initialize $w_0 = x_{pivot}$
2: **for** $i = 1, \ldots, N$ **do**
3:     Compute $g_i \leftarrow \zeta_i(w - x_i)/N$ for $w = x_{pivot}$ or $w_{i-1}$
4:     $\hat{g}_i \leftarrow Optimizer(g_i)$
5:     Let $w_i \leftarrow w_{i-1} - \eta_i\hat{g}_i$
6: **end for**
7: **return** $w_N$ (ensembled model)

---

simply letting $x_{pivot} = w_{i-1}$ at step $i$ of model ensembling. For instance, (4) becomes

$$
\begin{aligned}
w_1 &= x_{pivot} - \frac{1}{N}(x_{pivot} - x_1), \\
w_2 &= w_1 - \frac{1}{N}(w_1 - x_2), \\
&\vdots \\
w_N &= w_{N-1} - \frac{1}{N}(w_{N-1} - x_N),
\end{aligned}
\tag{5}
$$

where pivoted pseudogradients are now defined as $g_i = \zeta_i(w_{i-1} - x_i)/N$. Other choices (e.g., exponentially weighted moving average of $w_i$) are possible for $x_{pivot}$ to adaptively form pivoted pseudogradients. In particular, we may relate this form (5) back to the intuition provided in Section 1. That is, the pseudogradient is the difference vector between the current state of the ensemble $w$ and a model ingredient $x$, which takes advantage of the information signals of the training data already encoded in the weight space of $x$.

There is a vast and active literature on adaptive gradient algorithms, spanning both empirical and theoretical perspectives. Under the AME framework, this literature is now fully accessible to neural network weight-space aggregation, including convergence results and theoretical insights into optimizer selection for models demonstrating a range of characteristics. For more details, we refer to Appendix A.1.

## 4 Experiment Setups & Empirical Evaluation

**General Experiment Setup.** We examine the ensemble performance of multiple vision transformers in our evaluation. Per each model, a hyperparameter sweep was conducted whose grid contained a diversity of learning rates and adaptivity/momentum parameters, under different weight decay constants detailed in Appendix A.2. Given any training dataset, the resulting model for each hyperparameter configuration being swept over was saved after every training epoch. During the ensemble stage, all models were hierarchically listed based on their accuracy on a held-out validation set, conditioned on each epoch (e.g., at epoch 20). Afterwards, a select number of top-performing models were ensembled. For robust statistical significance at the expense of computational complexity, we repeated this ensemble procedure, including accessing new model ingredients per epoch, across all training epochs (typically ranging between 0 to 24 or 49). For example, each plot in Figure 2 contains the performance evaluation of approximately 1500 ensembles. We provide full details in Appendix A as well as in figure captions. For all experiments, we set the pseudogradient amplification schedule $\zeta_i = 1$, and employed adaptive pivoting to remain faithful to convex regularization against the stochastic objective (1). The pivots for AME are initialized at the standard model soup.

Figure 2 presents test accuracies of ViT-S ensembles trained on GLD-23K [37]. To assess whether AME constitutes a meaningful form of model averaging, we examine whether the resulting model preserves the memories of its model ingredients. Given that neural networks are known to memorize their training distributions [1, 2], a natural diagnostic is to compare performance on the original training distributions versus unseen test data–despite AME being applied without access to either. As shown in Appendix B, Figures 5-7 (extensions of Figure 2) and Table 2, the ensembled model

consistently achieves higher accuracy on training distributions than on test data, indicating knowledge retention of training data of the ingredients. Moreover, the smoothness of validation curves across a wide range of AME hyperparameters suggests that effective configurations generalize across training regimes: optimizer settings that ensemble well for weakly trained models also transfer to stronger ones.

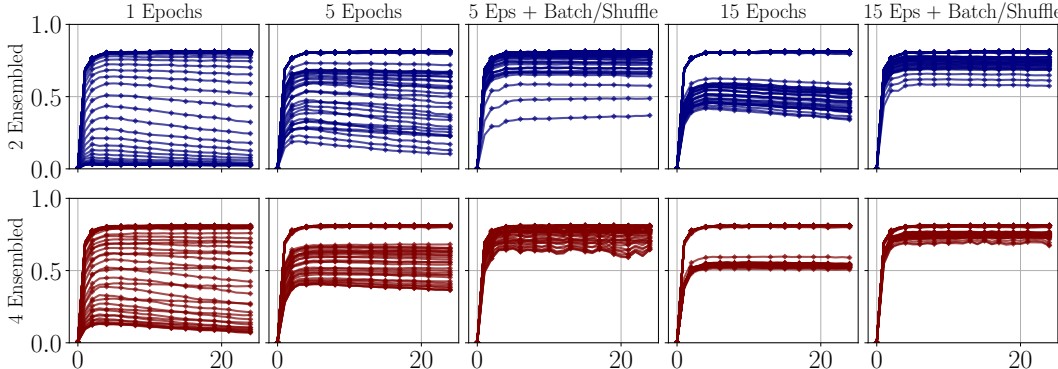

Figure 2: Ensemble test accuracy results of ViT-S fine-tuned on centralized GLD-23K dataset over hyperparameter sweep and experiment setup detailed in Appendix A.2. Each continuous line depicts a single hyperparameter configuration in AME, where an ensemble is formed for each x-axis timestep, which represents training epochs from 0 to 24. The title states the number of ensemble epochs where each model ingredient is treated as a datapoint. AME was instantiated with gradient descent, and batching used two ingredient models per batch. We observe that as the number of ensemble epochs range across 1, 5, and 15, *the ensemble performance improves and solidifies*, manifesting the effects of zero-data model training by simply increasing the number of ensemble epochs. Batching as well as shuffling the neural nets being fused further enhances the ensemble performance. This shows that simply running more meta-optimization epochs, with batching and shuffling of the ensemble ingredients, enhances performance. Additionally, adding more high-performance model ingredients also benefits test accuracy. Additional results are contained in Appendix B.

## 4.1   Extracting Domain Knowledge via Amortized Model Ensembling

It has previously been observed that balanced linear model averaging (model soup) may empirically strengthen generalization performance. If so, why might souping have this effect? We note that soups correspond to a single epoch of amortized model ensembling instantiated with GD, utilizing a harmonic learning rate schedule (Section 3). In order to capitalize on this phenomenon to design foundational meta-optimization strategies that attain greater generalization capacity, we present an experimental setup to isolate and manifest this behavior, navigated by the following heuristic.

**Intuition.**   During batched backpropagation, each minibatch gradient guides the model across a complex, non-convex optimization landscape, aiming to reach a weight configuration that aligns with the information contained in the minibatch sample. Through repeated updates over multiple batches, the objective is to converge to a local optimum that fits the training data distribution. However, once the model's weights have converged to such a configuration, the weights already encode the relevant information from the data. By computing a distance vector and adjusting the ensembled weights toward the converged model configurations, could these updates be interpreted as 'pseudogradients' toward the training data distribution? If so, this suggests that with suitably trained model weights and an appropriate optimizer, it is possible to achieve an effect analogous to model training while purely ensembling trained weights, without requiring any actual training data whatsoever. Is it possible to localize and prototype this phenomenon?

**Corresponding Setup.**   Motivated by this perspective, we examine the generalization capabilities of amortized ensembles by training ViT-S on CIFAR-10 while consecutively excluding selected classes for any given model. In the first setup, we trained 30 models with optimal AdamW hyperparameters and differing random seeds (impacting data batching and initialization of the classifier head), where the training dataloader singularly and evenly excluded each of the 10 classes. That is, all models were each trained on 9/10 classes, which forced an entire class of the training dataset (a highly problematic 10%) to be perfectly out-of-domain given any singular model. This definitively capped

testing time accuracy at $\sim 0.9$. Then, all 30 models were fused, and their performance evaluated. The analogous experiment was repeated at multiple even harsher extremes, where each model was exclusively trained on one to five classes, resulting in 5 sets of 30 models which were 90%, 80%, 70%, 60%, 50% out-of-domain given any singular model. Additional details about the setup are contained in Appendix A.2, and we summarize the final accuracies of the best performing singular model, model soup, and AME in Table 1. A detailed explanation of greedy model soups are presented in Appendix A.4, and supporting plots are provided in Appendix C.

Figure 10 in Appendix C confirms that as more epochs are taken toward the limited-class training data, the pre-trained transformer excessively overfits during fine-tuning, demonstrated by a destabilized test-time loss. However in Figure 11, the performance of well-performing ensembles steadily increase as more training epochs are taken (for a fixed number of ensemble epochs), indicating that each saturated training epoch is materializing a more informative ensemble ingredient in the model weight space. Despite the expectation of gross overfitting to the training classes in this setup, we observe that amortized model ensembling can significantly enhance OOD performance. Surprisingly, even in the case of 90%-OOD models, the best performing ensemble model reached a $\sim 23\%$ classification accuracy on the entire CIFAR-10 dataset. In this setting, each fine-tuned model is able to game the loss by learning a uniformly fixed mapping to the singular training class[3]. Additionally, Figures 11, 12 show phase transitions in test performance across AME hyperparameter configurations–a phenomenon we flag for future theoretical study (Appendix C).

Table 1: Test-time classification accuracy on CIFAR-10 dataset. For greedy soups, (Desc), (Asc) indicate the ordering of the model ingredients on a held-out validation set (Appendix A.4).

|  | **Best Model Ing.** | **Soup** | **Greedy Soup (Asc)** | **Greedy Soup (Desc)** | **AdamW Ensemble** |
|---|---|---|---|---|---|
| **10% OOD** | 87.99% | **98.81%** | 88.66% | 89.16% | **98.88%** |
| **50% OOD** | 49.65% | **96.64%** | 93.88% | 95.19% | **97.09%** |
| **60% OOD** | 39.78% | **94.88%** | 74.02% | 80.73% | **95.46%** |
| **70% OOD** | 29.89% | **84.77%** | 58.44% | 66.26% | **88.28%** |
| **80% OOD** | 20.00% | 61.18% | 58.32% | **62.91%** | **66.16%** |
| **90% OOD** | 10.00% | 21.76% | 22.44% | **22.45%** | **23.22%** |

Table 2: Train and test accuracies across different OOD percentages

|  | **90% OOD** | **80% OOD** | **70% OOD** | **60% OOD** | **50% OOD** | **10% OOD** |
|---|---|---|---|---|---|---|
| **Best Ingredient Train Accuracy** | 10.00% | 20.00% | 29.98% | 39.96% | 49.95% | 89.63% |
| **Best Ingredient Test Accuracy** | 10.00% | 20.00% | 29.89% | 39.78% | 49.65% | 87.99% |
| **Best Ensemble Train Accuracy** | **23.50%** | **66.85%** | **89.64%** | **96.75%** | **98.58%** | **99.99%** |
| **Best Ensemble Test Accuracy** | **23.22%** | **66.16%** | **88.28%** | **95.46%** | **97.09%** | **98.88%** |

**Discussion.** In Section 4.1, we materialize a setting in which ensembling can act to extract specialized domain knowledge, significantly enhancing the generalization capabilities of the ensemble net compared to any singular model or model soup. In Figure 2, we observe that simply employing more AME ensemble epochs helps to enhance, then stabilize the resultant ensemble performance for general hyperparameter selections, revealing the influence of implicit training during the zero-data ensembling process. Furthermore, the performance of the ensemble is higher on the seen training data than on the unseen testing data (e.g., Appendix B or Table 2). Notably, such training-like effects manifest despite AME being completely data-free, where regularization is performed against a convex objective (1). Additionally, instantiating the ensemble using various optimizer strategies can induce qualitatively differing behaviors in even a single ensemble epoch, as cumulatively illustrated in Figure 1. Due to limited space, we examine the effects of batching and shuffling while ensembling a different number of model ingredients in Appendix B. We relegate further discussions to the relevant portions of the Appendix.

## 4.2 Neural Averaging CIFAR-100

While our evaluations focus on vision classification with ViTs, the principles underlying AME are not tied to specific architectures. Indeed, the method's independence from training data and its general formulation suggest broad applicability. For instance, what does it mean to perform a neural average over training data? As an illustrative example, consider reinterpreting input images as weights in an

---

[3]Indeed in Figure 10, the training accuracy achieves 100.00% within the first three epochs.

augmented architecture: each image is treated as defining the weights of an added initial layer, and a synthetic input consisting entirely of the pixel value 1 is passed through this modified network. All resulting models thus recover the original image classification under the identity data. Therefore, each image corresponds to a unique network, and ensembling the first-layer weights becomes equivalent to ensembling the image data itself.

Motivated by this intuition, we construct a synthetic dataset by sampling 500 image tensors initialized over a uniform distribution $U[0, 1]$, matching the number of training examples per class in CIFAR-100. We then apply class-conditional AME, treating each training image as an ensemble ingredient. The generated pseudo-dataset still elicits correct predictions on the classes the images were synthesized from (e.g., "willow tree (95%)" vs. "maple tree (82%)") from a typical ViT fine-tuned on CIFAR-100. This suggests that AME captures meaningful structure in weight space that persists even under input synthesis, raising fundamental questions about the nature of neural averaging. We provide additional visualizations in Appendix E.

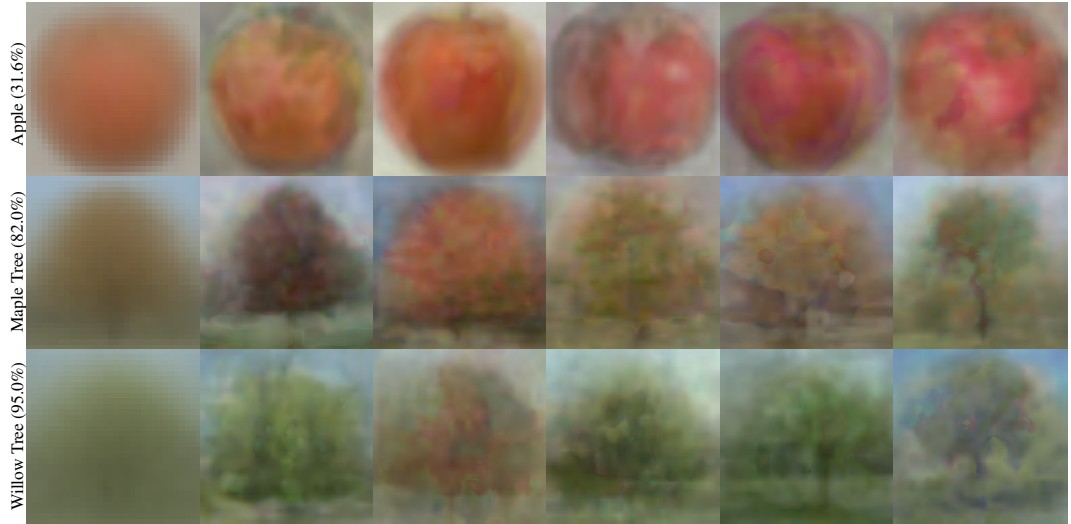

Figure 3: The first column displays uniformly averaged (souped) images from each CIFAR-100 class's training set. Subsequent columns show analogous outputs from AME, applied to 500 randomly initialized image tensors per class. Image synthesis was performed using AdamW-based AME with 40 optimization sweeps. The classifier–a ViT fine-tuned solely on the original CIFAR-100 dataset–was never exposed to the synthesized data. Percentages indicate the proportion of synthesized images per class correctly classified by the ViT (out of 500), where random-guessing accuracy is 1%. For further visualizations and AME hyperparameter details, we refer to Appendix E.

## 4.3 Neural Optimizers as General Algorithms for Online Statistical Estimation

We now propose an alternative interpretation of deep learning optimizers: *optimizers can be viewed as streaming algorithms for computing statistical estimators*. From this perspective, each update involves computing a distance vector between the current estimator and a new data point (or minibatch), adjusted according to rules defined by the optimizer. In AME, we have interpreted these objects as *pseudogradients*, but this principle applies more generally for any type of problem convertible into an online inference setup. This framework naturally accommodates batching and allows us to define an entire family of estimators parameterized by the optimizer's hyperparameters. In the experiments below, we validate this interpretation empirically and show that AME can outperform the empirical average, especially in heavy-tailed or noisy regimes–linking back to the discussion in Section 3.1. A more in-depth explanation with additional details is given in Appendix F.

**Synthetic Experiment Setup.** We draw 60,000 samples from a heavy-tailed Cauchy distribution and treat this as the 'population' from which observations can be drawn. From this, we subsample 300 points to simulate the ensemble ingredients, analogous to materializing 300 two-dimensional model weights. We then compute both the model soup average and the AME-ensembled result (using

Adam). This trial is repeated 300 times, and the results are visualized in Figure 4 (a-b), where (c) provides the result for a standard Gaussian distribution.

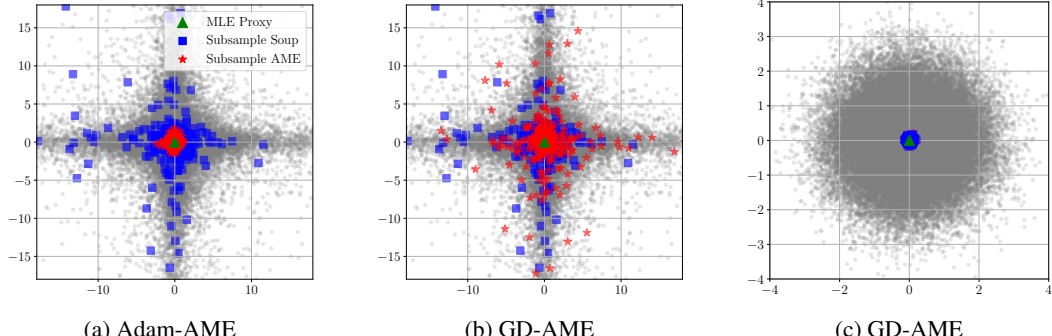

|   (a) Adam-AME   |   (b) GD-AME   |   (c) GD-AME   |

Figure 4: Each panel compares AME ensembles (Adam or GD) against model soup using identical data samples per trial. In (a-b), we verify that under the heavy-tailed Cauchy distribution, Adam-AME demonstrates superior performance than GD-AME. For a non-heavy-tailed Gaussian (c), GD-AME ensembles (red) align closely with the soups (blue) centered around the green MLE, and are therefore visually occluded. Full details and hyperparameter settings are described in Appendix F.

This reveals that model soup exhibits high variance and completely fails to converge toward the proxy MLE under subsampling in heavy-tailed regimes. In contrast, Adam-AME produces estimates that diverge from the empirical average but reliably converge near the proxy MLE. By contrast, GD-AME, which necessarily must only linearly interpolate between the model ingredients, demonstrates similar properties to model soup estimators. This confirms that AME functions as an optimizer-based statistical estimator rather than a simple average, and can yield improved performance depending on the underlying distribution when instantiated with an adaptive optimizer.

The central goal of AME is to harness insights from optimization theory to construct a novel class of empirical estimators that go beyond simple linear averaging. Rather than merely approximating the empirical mean, AME leverages existing optimizers to synthesize ensemble models that encode richer statistical structure–structures often missed by methods such as model soup. This reinterpretation positions adaptive optimizers as numerical routines for computing robust, noise-aware estimators. By initializing at a fixed point and applying pseudogradient updates toward sampled models or datapoints, AME produces solutions that converge closer to the population mean than the empirical average. We believe this perspective motivates the development of future deep neural network optimization-inspired ensembling techniques as statistical estimators in their own right.

## 5 Conclusion

What does it mean to perform a neural average–to distill a collection of models into a single network that faithfully reflects their collective knowledge? A principled formulation need not rely on access to any data during the averaging process, yet the resulting model should inherit properties of its ingredients, performing better on their respective training distributions than on unseen data despite never observing either. Drawing inspiration from model soup, we develop a notion of neural averaging through the idea of pseudogradients: directions in parameter space that implicitly encode minibatch information. While prior work interprets the out-of-domain gains of model soup through Bayesian lenses (e.g., maximum likelihood estimation), we offer a complementary optimization-based perspective: uniform model averaging is equivalent to a single epoch of gradient descent under a harmonic learning rate schedule. Extending this view, we demonstrate that adding meta-optimization epochs, batching models, and incorporating adaptive optimizers yields consistent performance gains. These insights culminate in the Amortized Model Ensembling (AME) framework, which reframes weight-space averaging as zero-data meta-optimization. Our formulation sheds light on the generalization behavior of weight-averaged deep networks and defines, in one sense, how to perform neural averaging.

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

# A  Supplementary Material, Experiment Setups, and Datasets

## A.1  Advantages of Amortized Model Ensembling as Meta-Optimization

There is a vast and active literature on adaptive gradient algorithms, spanning both empirical and theoretical perspectives. Under the AME framework, this literature is now fully accessible to neural network weight-space aggregation, including convergence results and theoretical insights into optimizer selection for models demonstrating a range of characteristics. For instance, under appropriate conditions, we may show that AME can do no worse than linear ensembling or linear weight interpolation. We also have the following very well-known heuristic:

**Proposition 1.** *(Informal) There exists a learning rate schedule $\eta_i$ and adaptivity parameter schedule $\varepsilon_i$ such that **Adagrad Ensembling$\approx$ GD Ensembling**.*

*Proof.* At each step $i$, let $\eta_i = \widetilde{\eta}_i \varepsilon_i \gg \sum_{j=1}^i g_j^2$. Then, we have

$$w_i = w_{i-1} - \frac{\eta_i}{\sqrt{\sum_{j=1}^i g_j^2} + \varepsilon_i} g_i \approx w_{i-1} - \widetilde{\eta}_i g_i,$$

where the rightmost side corresponds to souping via standard linear averaging, for learning rate and amplification schedules $\widetilde{\eta}_i = \zeta_i = 1$. $\qquad\square$

In summary, this heuristic follows from the observation that in adaptive optimizers, adaptivity may effectively be turned off by selecting a large adaptivity parameter, then negating this effect by applying a learning rate of identical magnitude, scaled multiplicatively by $\mathcal{O}(1)$. Analogous results hold for **Adam Ensembling**, with appropriate momentum parameters. Appendix A.3 contains more formal convergence results for AME.

**Customized model aggregation strategies.**  In Section 3.1, we saw that the previous paradigm of taking a linear model average is equivalent to non-adaptive gradient descent over a convex online objective with optimum centered at $\xi \sim \mathcal{D}$. Of particular interest is the case in which the distribution of $\mathcal{D}$ is heavy-tailed, where gradient descent is known to have significant convergence difficulties [10, 31, 38, 39]. Recent works have suggested a relationship between heavy-tailed stochastic gradients and the superior performance of Adam over SGD [11, 33], which intuits that Adam ensembling may be more advantageous than linear average aggregation if the distribution $\mathcal{D}$ of model ingredients is heavy-tailed. Furthermore, there are numerous adaptive or clipped optimizers developed within the literature to mitigate diverse challenges that arise over the optimization landscape [9, 11, 40, 41, 42]. Each optimizer may be used to ensemble model weights within our framework, which is optimizer-agnostic.

**Compute Efficiency.**  Each ensembling step carries materially less computation complexity than a single adaptive gradient descent step during model training. Additionally, we note that our algorithm operates independently from coordinate correlation; that is, the weights of each layer can be loaded separately, then fused, to provide the weights of the output model. Therefore, we entirely sidestep the memory/computation bottleneck as ensembling may be applied to arbitrary partitions of model weights.

## A.2  Experiment Setups

Data batch size was fixed to 64 during model training, and the loss was set to cross entropy. After conducting initial experiments comparing optimization dynamics under AdamW, Adagrad, Adadelta, and SGD, AdamW was selected to train all models. When any given optimizer was deployed during model training or at the ensemble stage, we consistently swept over the hyperparameter grid given in Table 3, where $\lambda$ represents the weight decay constant, $\eta$ the learning rate, $\varepsilon$ the adaptivity parameter or smoothing term, and $\beta/\rho$ the momentum parameters.

**Dataset Setup.**  We mainly utilized two datasets in the empirical evaluation as follows.

**CIFAR-10**: The CIFAR-10 dataset [43] is composed of 60,000 color images of size $32 \times 32 \times 3$, split into 10 classes. There are 50,000 training images and 10,000 test images, with labels

Table 3: Optimizer Hyperparameter Configurations

| GD | Adagrad |
|---|---|
| $\lambda \in \{0.0, 0.1\}$, $\eta \in \{\text{np.linspace}(10^{-7}, 2, 30)\}$ | $\lambda \in \{0.0, 0.1\}$, $\eta \in \{0.0001, 0.001, 0.01, 0.1\}$, $\varepsilon \in \{10^{-9}, 10^{-7}, 10^{-5}, 10^{-3}\}$, $\beta_1 \in \{0.9\}$ |
| **AdamW** | **Adadelta** |
| $\lambda \in \{0.0, 0.01, 0.1\}$, $\eta \in \{0.0001, 0.001, 0.01, 0.1\}$, $\varepsilon \in \{10^{-9}, 10^{-7}, 10^{-5}\}$, $\beta_1 \in \{0.8, 0.9\}$, $\beta_2 \in \{0.9, 0.999\}$ | $\lambda \in \{0.0, 0.1\}$, $\eta \in \{0.01, 0.1, 1, 2\}$, $\varepsilon \in \{10^{-8}, 10^{-6}, 10^{-4}\}$, $\rho \in \{0.8, 0.9, 0.999\}$ |

representing objects such as airplanes, automobiles, birds, cats, and dogs. CIFAR-10 is widely used for benchmarking deep learning models, especially convolutional neural networks (CNNs).

**GLD-23K**: The GLD-23K dataset is a subset of the GLD-160K dataset introduced in [37]. It contains 23,080 training images, 203 landmark labels, and 233 clients. Compared to CIFAR-10, the landmarks dataset consists of images of far higher quality and resolution, and therefore represents a more challenging learning task. Originally, GLD-23K is a real-world high-quality benchmark for federated learning algorithms. Due to the quality and resolution of its images, we converted this decentralized dataset into a central setting to evaluate ensemble performance, where the training/testing distributions are preserved.

**General Setup.** In this work, we investigate the performance of amortized model ensembling using foundational image classification architectures. To rigorously assess the impact of ensembling on performance at the expense of memory/computational complexity, we saved all model weights after every single training epoch, typically across a range of 0 to 24 epochs, for each hyperparameter configuration in the grid. During the ensembling stage, again at each of the epochs between 0 to 24, we followed the procedure below.

**(1)** Evaluate the validation performance of all trained models for a fixed dataset and architecture, for the hyperparameter sweep detailed in Table 3.

**(2)** Rank the models hierarchically based on the validation metric, from best to worst or vice versa.

**(3)** Select the top $n$ models for ensembling.

After each descent step against the objective (1), we decayed the learning rate harmonically (unless explicitly stated otherwise in the figure caption). To confer high statistical validity, this procedure was repeated for every epoch between 0 and 24, for all hyperparameter configurations listed in Table 3 during the amortized ensembling phase. Per each ensemble conditional on a training epoch (e.g., 21) and ensemble optimizer hyperparameter configuration, the ensemble performance was evaluated. Therefore, most figures in the paper, including those in the Appendix, each showcase the performance of at least tens of thousands of ensembled models across these settings.

**Setup for Table 1.** We now describe in more detail the dataset partitioning and exclusion within the experiment setup. 6 different scenarios were evaluated, corresponding to 10%, 50%, 60%, 70%, 80%, 90% out-of-domain data given any singular model. Each setting shared a similar training structure, where ten dataloaders were instantiated, ranging from class 0 to class 9. In the case of 10% OOD, the class $n$ dataloader excluded all training data from CIFAR class $n$. In the case of 50% OOD, the class $n$ dataloader excluded all datapoints of classes $n, n+1, \ldots, n+4$, where the excluded class was calculated up to modulo 10. The other extremes followed an identical setup; for instance, the class 1 dataloader for the 80%-OOD setting trained on only classes 0 and 9 from the CIFAR-10 training data. Each ViT was optimized via the best performing AdamW hyperparameter configuration discovered during the grid search in Table 3, which corresponded to $\eta = 1e\text{-}4$, $\varepsilon = 1e\text{-}5$, $\beta_1 = 0.8$, $\beta_2 = 0.9$,

and $\lambda = 0.0$. Each individual model per OOD setting used a unique Pytorch and dataloader generator random seed combination, impacting the initialization of the 10-class linear probe classifier as well as dataloader batch ordering. In total, 30 ensemble ingredients were uniformly trained, 3 optimized on each class dataloader in the manner detailed above.

### A.3   Convergence Results for Amortized Model Ensembling

Recasting model weight-space averaging as optimization renders classical convergence results applicable to amortized model ensembling. This implies the practicability of a wealth of meta-optimization strategies and heuristics that draw from well-established literature in adaptive or clipped optimization. For instance, we present the following proposition.

**Proposition 2.** *Let $\xi_i \overset{iid}{\sim} \mathcal{D}_{train}$ for $\|\mathbb{E}[\xi]\| < \infty$, where $i \in \{1, \ldots, N\}$ and $\mathcal{D}_{train}$ is the (possibly mixture) distribution of trained model weights, where the pseudogradient amplification schedule is bounded, $|\zeta_t| \leq \zeta$. Then, the following statements hold.*

*(i) Given a fixed learning rate schedule, quadratic ensembles instantiated under gradient descent need not converge.*

*(ii) Let the learning rate decay schedule satisfy $\eta_t = \min\{\mathcal{O}(t^\alpha), 1/(2 \operatorname{diam}(\mathcal{X}\zeta)\}$ and $\alpha < -1$, where $t$ is the number of backpropagation timesteps. Here, $\mathcal{X}$ is a closed ball centered at the origin which subsumes the $N$ ensemble ingredients as well as the initialization. Then, amortized ensembling converges under GD, Adagrad optimizer strategies, where convergence is also attained for Adam for $\beta_1^2 < \beta_2$.*

*(iii) Model soups converge in probability to the first moment of $\mathcal{D}_{train}$, that is,*

$$\lim_{n \to \infty} \mathbb{P}\left(|x_{soup,n} - \mathbb{E}[\xi]| < \varepsilon\right) = 1$$

*for any $\varepsilon > 0$.*

*Proof.* Firstly, note that the proof of **(iii)** is clear by the weak law of large numbers. It remains to prove the other two propositions, which is straightforward.

Proof of **(i)**. We provide a class of counterexamples via explicit construction, which can be easily generalized to higher dimensions. For readability and ease of notation, we work in the $\mathbb{R}^2$ plane. For $k, \omega \in \mathbb{R}_{>0}$, set $\eta = 1/(k+1)$ where the $N = 4$ ensemble ingredients are given by

$$x_1 = (-k\omega, (k+1)\omega), \quad x_2 = (-(k+1)\omega, -k\omega), \quad x_3 = (k\omega, -(k+1)\omega), \quad x_4 = ((k+1)\omega, k\omega).$$

Then, it is clear that initialization on $(\omega, 0)$ induces perpetual cyclic dynamics on the $\ell_1$ unit ball.

Proof of **(ii)**. We now show that decaying the learning rate as above must guarantee convergence for an arbitrary set of ensemble ingredients. As $N < \infty$, there must exist a closed ball $\mathcal{X}$ such that $\{\xi_0, x_1, \ldots, x_N\} \subset \mathcal{X}$ where $\xi_0$ is the initialization of the ensemble. This implies that the tail ends of GD ensemble updates must decay, that is,

$$A_n = \left\|\sum_{t=n}^{\infty} \frac{\eta_t \zeta_t (x_t - \xi_t)}{N}\right\| \leq \sum_{t=n}^{\infty} \eta_t \frac{\zeta \operatorname{diam}(\mathcal{X})}{N} \approx \int_n^{\infty} \mathcal{O}(t^\alpha) \, \mathrm{d}t \to 0$$

as $n \to \infty$, where $\xi_t$ is the intermediary ensemble at step $t$. Here, we used that the maximum distance between any two points within $\mathcal{X}$ is $2 \operatorname{diam}(\mathcal{X})$, and that $\eta_t \cdot 2 \operatorname{diam}(\mathcal{X})\zeta < 1$ implies that the ensemble $\xi_t$ remains within the convex set $\mathcal{X}$. In the case of Adagrad, we have for $g_i$ the amplified pseudogradient updates

$$B_n = \left\|\sum_{t=n}^{\infty} \frac{\eta_t \zeta_t (x_t - \xi_t)}{N\left(\sqrt{\sum_{i=1}^t g_i^2} + \varepsilon\right)}\right\| \lesssim \sum_{t=n}^{\infty} \eta_t \to 0$$

as above. In short, this result comes from the automatic clipping of the update lengths that occurs via adaptivity. Similarly for Adam, we project the model after every update to the closed ball $\mathcal{X}$. We

then have

$$
C_n = \left| \sum_{t=n}^{\infty} \eta_t \frac{1}{1-\beta_1^t} \cdot \frac{\beta_1^t m_0 + (1-\beta_1)\sum_{r=1}^{t-1} \beta_1^{t-r} g_r + (1-\beta_1)g_t}{(\sqrt{1-\beta_2^t})^{-1}\sqrt{\beta_2^t v_0 + (1-\beta_2)\sum_{r=1}^{t-1}\beta_2^{t-r}g_r^2 + (1-\beta_2)g_t^2} + \varepsilon} \right|
$$

$$
\leq \sum_{t=n}^{\infty} \eta_t \frac{\sqrt{1-\beta_2^t}}{1-\beta_1^t} \left| \frac{\beta_1^t m_0 + (1-\beta_1)\sum_{r=1}^{t-1}\beta_1^{t-r}g_r + (1-\beta_1)g_t}{\sqrt{\beta_2^t v_0 + (1-\beta_2)\sum_{r=1}^{t-1}\beta_2^{t-r}g_r^2 + (1-\beta_2)g_t^2}} \right|
$$

$$
\leq \sum_{t=n}^{\infty} \eta_t \frac{\sqrt{1-\beta_2^t}}{1-\beta_1^t} \sqrt{\frac{\beta_1^{2t}m_0^2}{\beta_2^t v_0} + \frac{(1-\beta_1)^2}{1-\beta_2}\sum_{r=1}^{t-1}\frac{\beta_1^{2(t-r)}}{\beta_2^{t-r}} + \frac{(1-\beta_1)^2}{1-\beta_2}}
$$

$$
\lessapprox \int_{t=n}^{\infty} \mathcal{O}(t^{\alpha})\sqrt{\frac{m_0^2}{v_0} + \frac{(1-\beta_1)^2}{(1-\beta_2)}\frac{\beta_1^2}{\beta_2 - \beta_1^2}}\, dt < \infty.
$$

We note that $\beta_1^2 < \beta_2$ often works well in practice, e.g., $\beta_1 = 0.8$ or $0.9$ and $\beta_2 = 0.99$. $\qquad\square$

### A.4  On Greedy Soups and Greedy Model Ensembling

**Remark on Greedy Ensembles.**   We may generalize the greedy souping approach given in [3] to greedy amortized ensembling. In prior works [4, 44], greedy aggregation refers to the paradigm of ordered, iterative model averaging, where each newly formed average is kept if and only if it demonstrates enhanced performance on a held-out validation set. This often provides a marginal, yet statistically significant improvement in the resulting model performance. Analogously, in greedy amortized ensembling, we reject a new ensemble at each step and revert to the immediately preceding model/optimizer state when performance enhancement was not detected. However, we note that our definition of 'greedy' may differ from those of some papers (e.g., [3, 45]), which often may implicitly initiate the averaging process from the best-performing model ingredient. Such a setup definitionally ensures that the end result can be no worse than the pre-existing optimal, fully-trained model, even if all other ingredients are greedily dropped.

Therefore, in our work, we evaluated whether a greedy strategy can indeed enhance model comprehension or generalization capabilities, or whether it leverages statistical tautologies such as a lower bound on the accuracy by starting with the best-performing trained model. In our greedy ensembling approach, consistent with the interpretation of pseudo-gradient descent in Section 3.3, we started with the weakest performing model out of the ensemble ingredients and employ a greedy ensemble strategy. We observed that by hastily rejecting models due to not immediately attaining marginal improvements on the held-out validation set, a greedy framework often performed significantly worse than a fairer, all-encompassing ensemble strategy which does not neglect to ensemble weights based on ephemeral or immediate held-out performance. Therefore, these results were generally not included in the paper, due to performing *worse* than the standard ensemble. However, we included the performance of the greedy ensemble initialized at the best, as well as worst performing models in the setting of Section 4.1. Table 1 indicates that the ordering of the ingredients matters for greedy souping; starting from the strong model on a held-out set, then adding weaker ingredients sequentially tends to dominate over starting with the worst performing model and then adding stronger ones.

# B  Gradient Descent Ensembles for GLD-23K

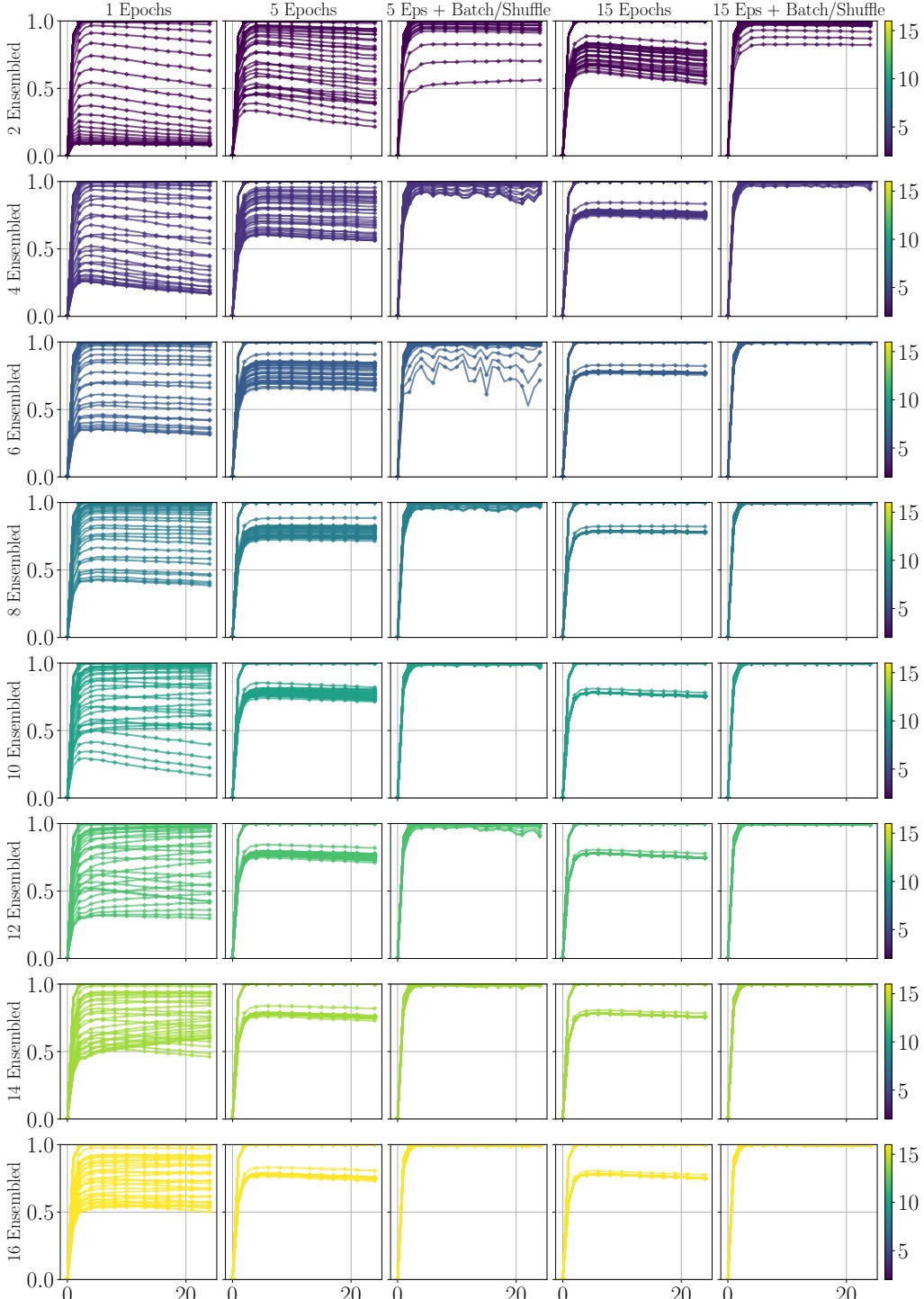

Figure 5: ViT-S ensemble performance on GLD-23K training data (training accuracy), where the x-axis for each plot is a training epoch. The title states the number of ensemble epochs used, per each training epoch ranging from 0 to 24. The y-axis is set to $(0, 1)$. Compared to Figure 6, which gives analogous results for a held-out validation set, the best performing ensembles reach near 1 accuracy. It can be seen that additional ensemble epochs as well as Batching/Shuffling can greatly assist in enhancing ensemble performance. All batching used two ingredient models per batch in this section.

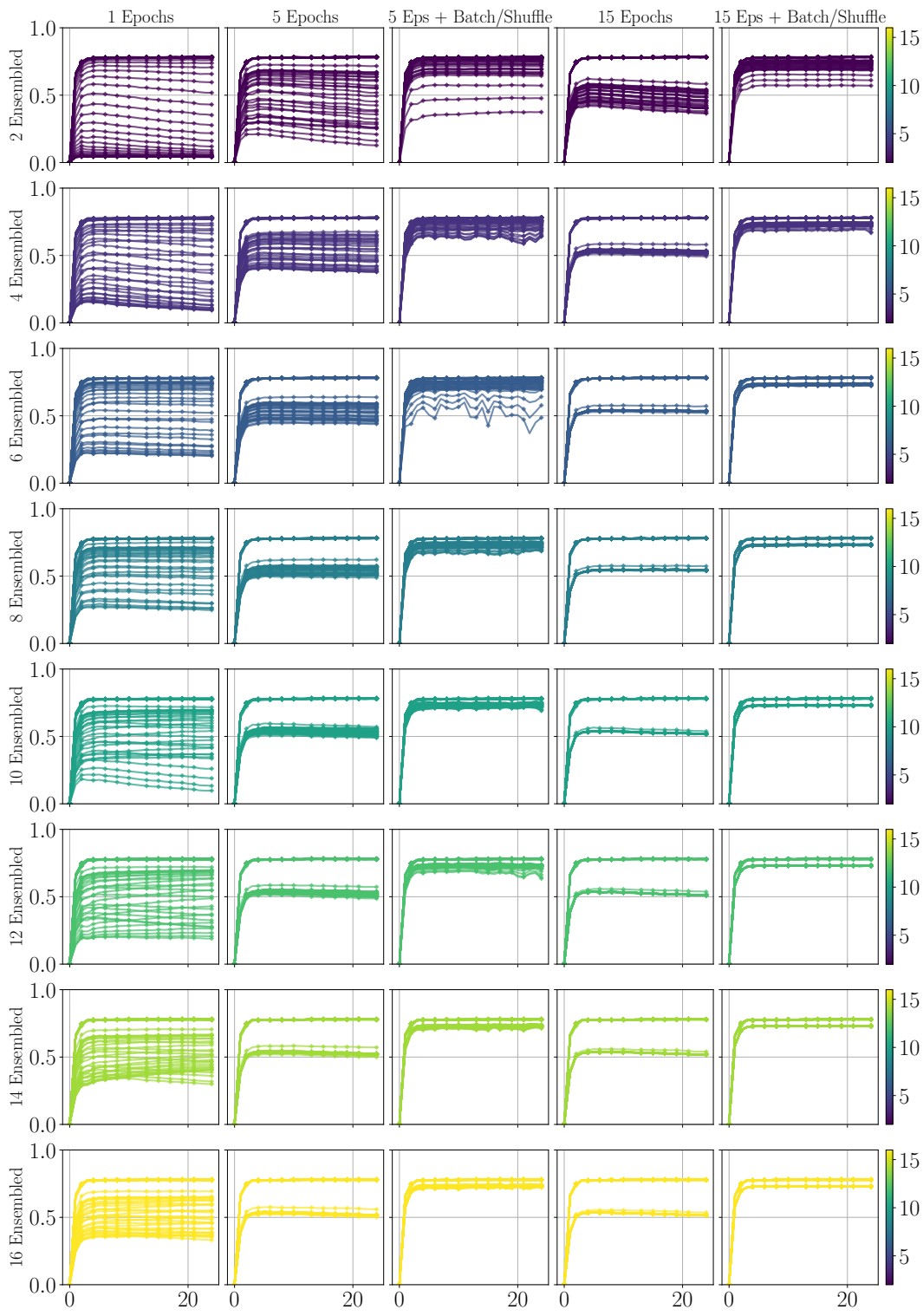

Figure 6: Analogous setup to Figure 5, where the validation accuracy is plotted. The validation set was disjointly partitioned from the GLD-23K dataset training distribution, and unseen during training. $y$-axis is artificially fixed to $(0, 1)$ for clearer comparison. We note the similarities in performance on the GLD testing distribution, given in Figure 7.

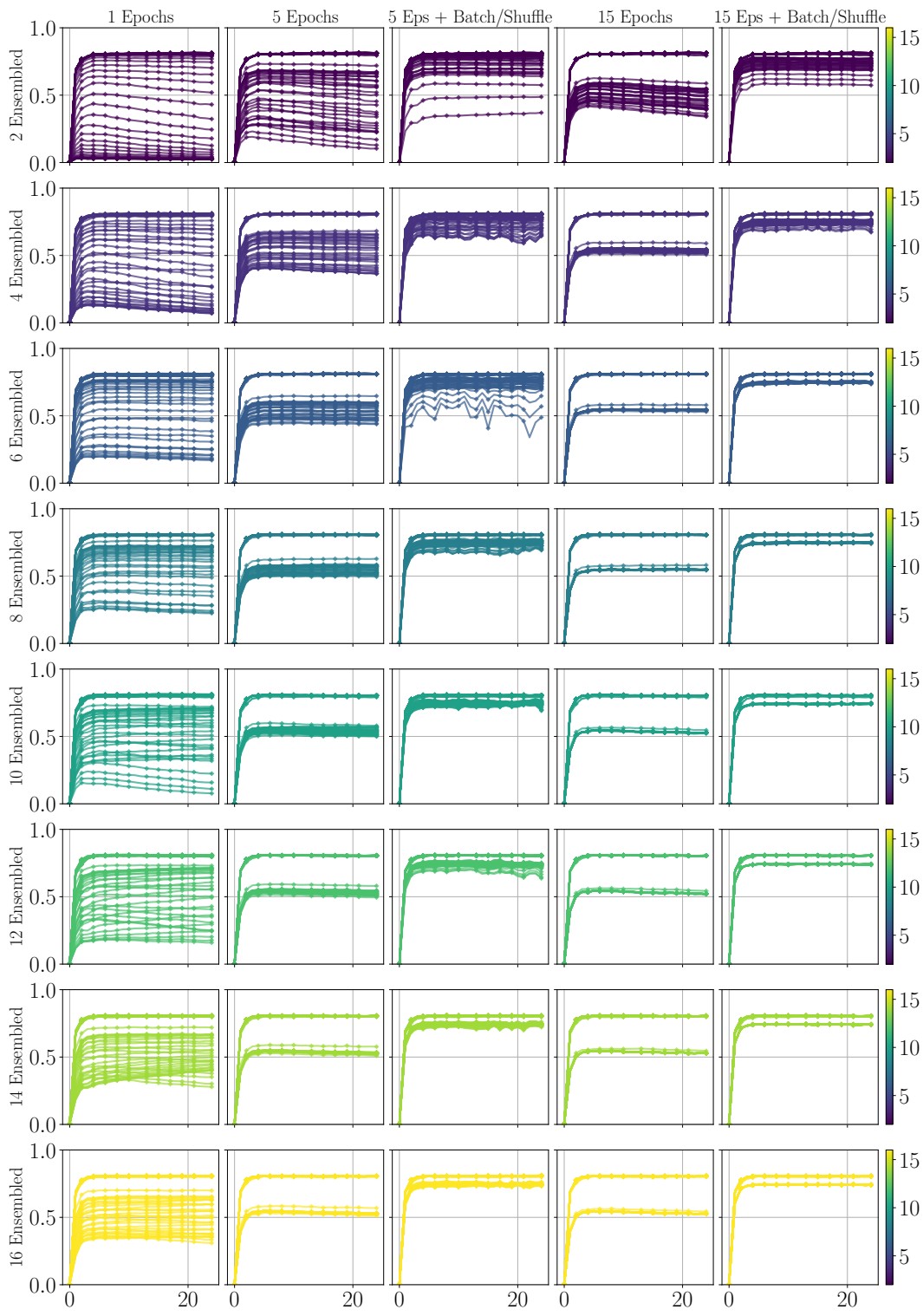

Figure 7: This figure provides the counterpart to Figure 6, where accuracy is computed across the testing distribution rather than a held-out validation distribution. Both testing and validation distributions attain comparatively reduced performance in the ensemble, compared to the training distribution which was seen during fine-tuning (Figure 5).

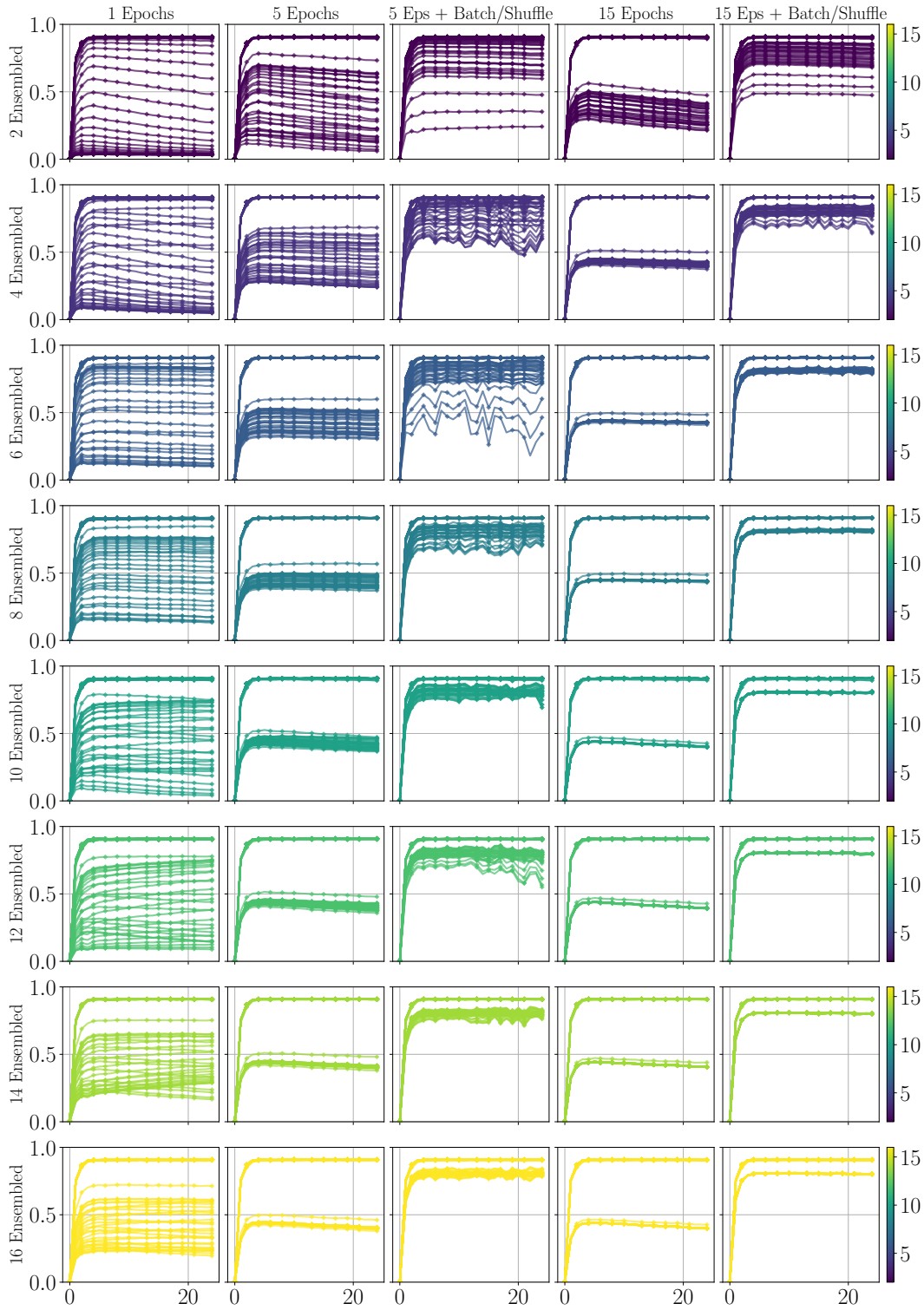

Figure 8: In Figures 8, 9, the entire GLD-23K dataset is adulterated with Gaussian noise with standard deviation 1/4 [46, 47, 48]. The accuracy is plotted in Figure 8, across the entire noised dataset to evaluate for robustness to covariate shift. The y-axis is artificially set to $(0, 1)$ for easier comparison, and no significant performance degradation is observed. We observe that even under this adversarial setting, deploying more ensemble epochs as well as batching/shuffling confers significant adversarial robustness, confirming zero-data training-like effects during amortized model ensembling.

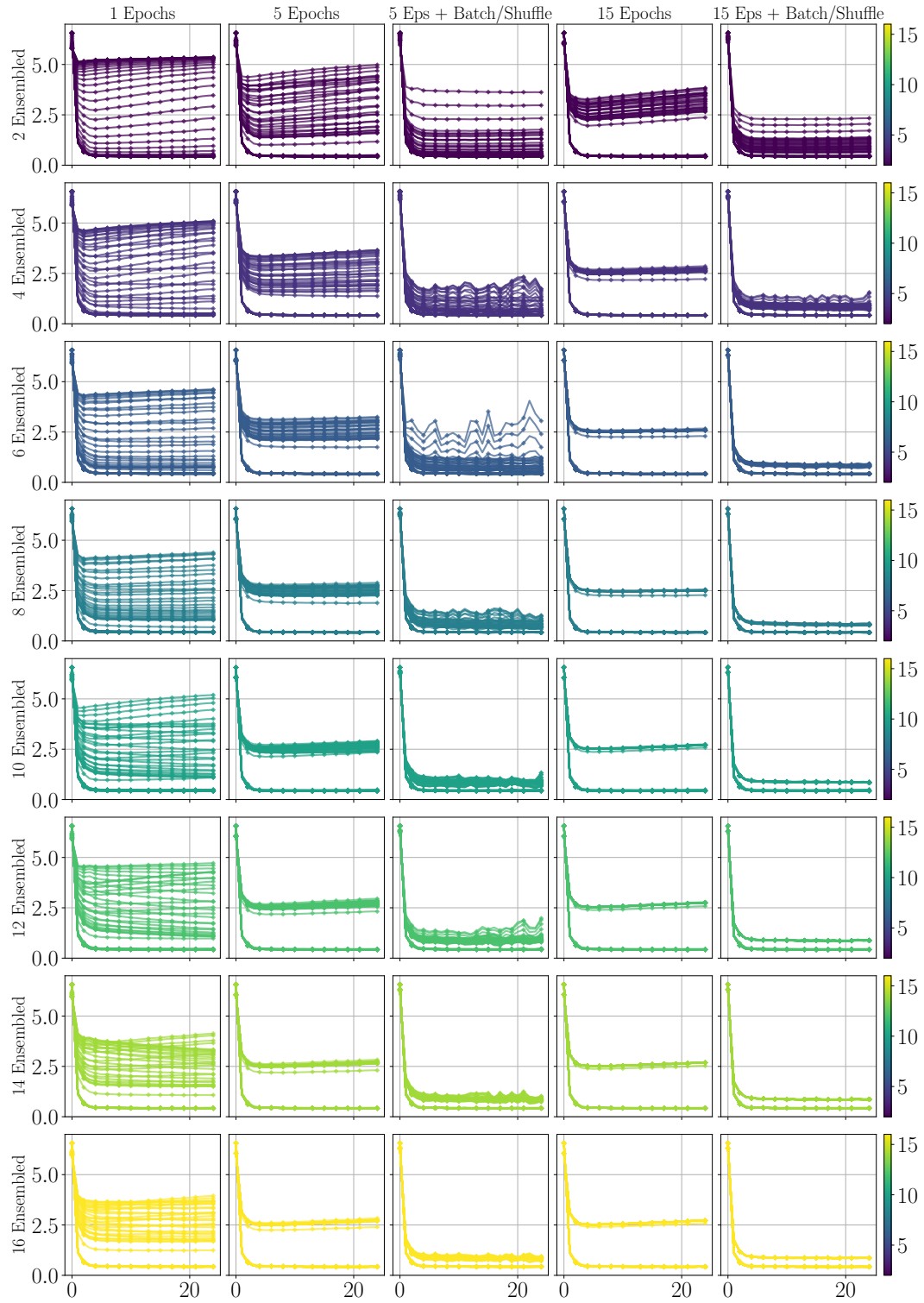

Figure 9: We use the identical setup as in Figure 8, where the loss is plotted instead of accuracy. We observe that higher accuracy performance of quadratic ensembles are visible as being inversely proportional to loss, indicating that ensembles are calibrated. This is in contrast to model soups, where miscalibration is a noted issue [3]. The y-axis is fixed to $(0, 7)$ for better visualization.

## C Supporting Plots for Out-of-Domain Amortized Ensembling (Section 4.1)

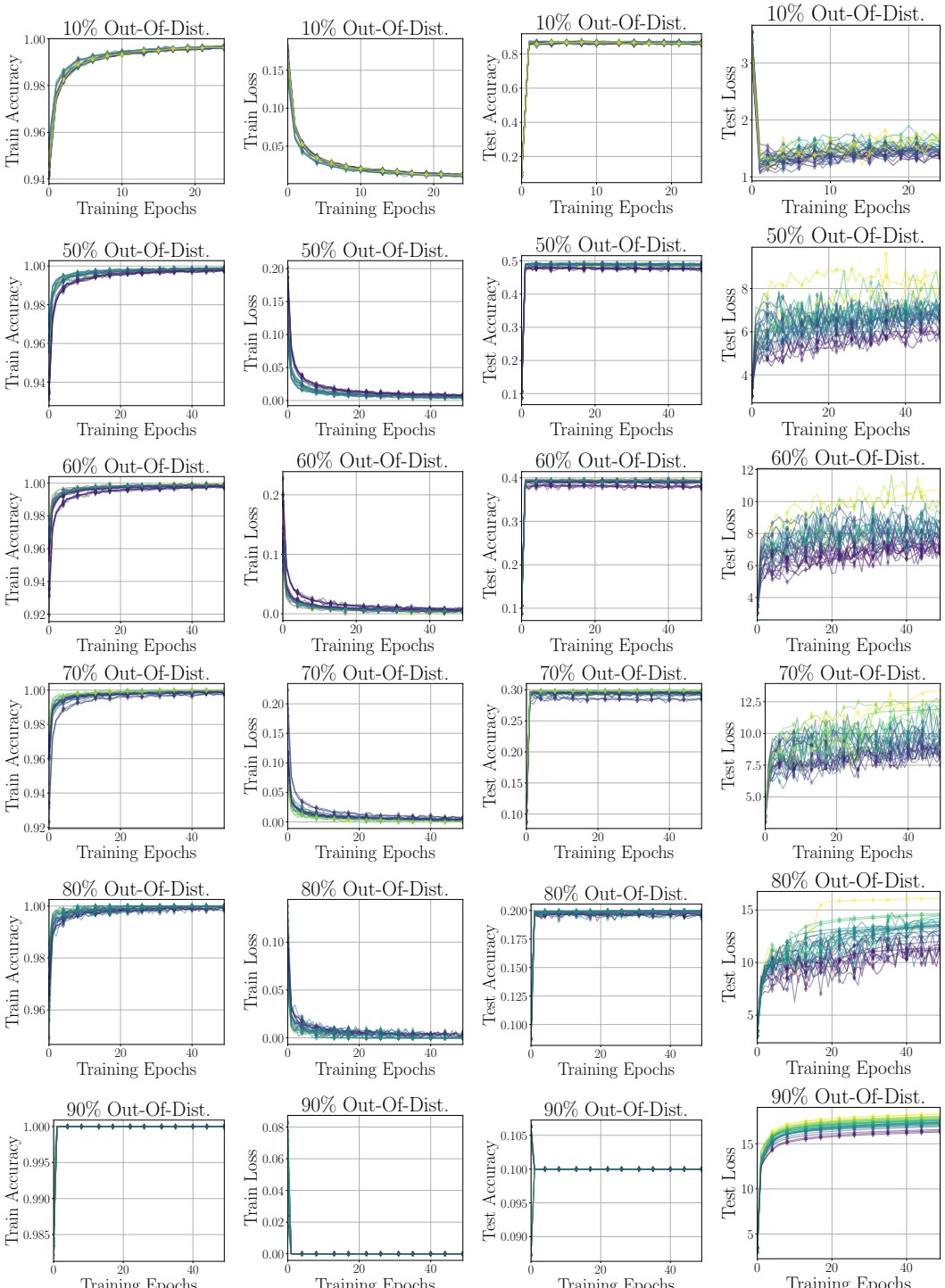

Figure 10: Train/Test performance for model ingredients, where time series data are organized hierarchically based on the final test loss at epoch 50, before applying a colormap with a gradient transitioning from dark to light. Lighter colors correspond to higher test loss values. The test loss for all OOD settings (rightmost column), evaluated on the entire CIFAR-10 test data partition, increases with each additional training epoch, indicating overfitting to the training data. Due to viewing limited classes during training, each model ingredient is overconfident in their capabilities (leftmost column).

Over a sweep of optimizer hyperparameter configurations, the time series accuracy data for AME ensembles appear to separate into four distinct phases as training epochs progress: (a) low loss with high accuracy, (b) low loss with unstable accuracy, (c) medium loss with unstable accuracy, similar to that observed in model soup, and (d) high loss with model soup-level accuracy. Notably, prior work has indicated that model soup exists in such a miscalibrated regime [3]. Typically, the test-time ensemble loss splits into three distinct branches clearly visible in the color coding over varying hyperparameter configurations, a pattern also observed in settings with lower percentages of OOD data (Figure 12). We leave an explanation of this curious phenomena for future research.

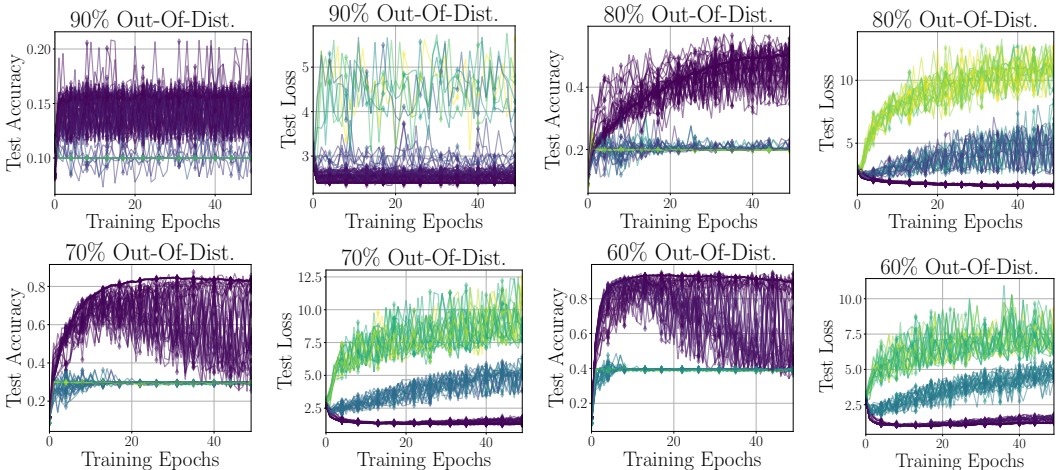

Figure 11: Models are ensembled for 35 epochs using a batch size of 15. To assess calibration, the test accuracies are hierarchically organized by final test loss and visualized using color coding, where darker colors represent lower test loss. Test accuracy is plotted next to corresponding test loss, sharing the identical color coding. Intriguingly, the worst performing ensembles with respect to test loss for the 70%-OOD setting attain $\sim 30.00\%$ test accuracy, which has significantly higher loss than the worst performing ensembles for the 90%-OOD setting with $\sim 10.00\%$ test accuracy.

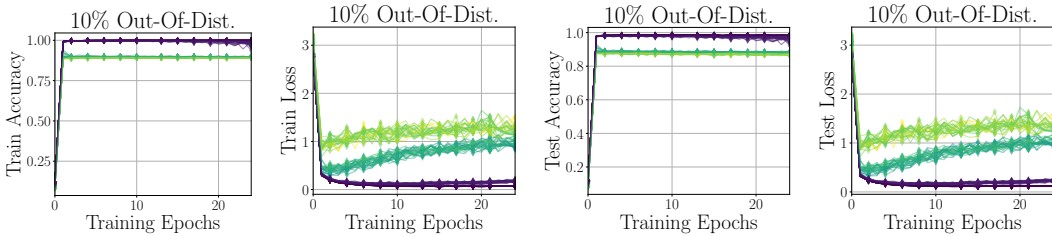

Figure 12: Ensemble performance over 25 training epochs. We note that the best performing training and testing accuracies are 99.99 % vs 98.88 %, where a higher accuracy is attained for data seen by the ensemble ingredients during training. Similar trends are observed for other ensembles (Table 4), which strongly suggests that the enhanced generalization performance of model soups are attained via zero-data meta-training during the single epoch of gradient descent when viewed as GD instantiation of AME.

Table 4: Train and test accuracies across different OOD percentages

|  | 90% OOD | 80% OOD | 70% OOD | 60% OOD | 50% OOD | 10% OOD |
|---|---|---|---|---|---|---|
| **Best Ingredient Train Accuracy** | 10.00% | 20.00% | 29.98% | 39.96% | 49.95% | 89.63% |
| **Best Ingredient Test Accuracy** | 10.00% | 20.00% | 29.89% | 39.78% | 49.65% | 87.99% |
| **Best Ensemble Train Accuracy** | 23.50% | 66.85% | 89.64% | 96.75% | 98.58% | 99.99% |
| **Best Ensemble Test Accuracy** | 23.22% | 66.16% | 88.28% | 95.46% | 97.09% | 98.88% |

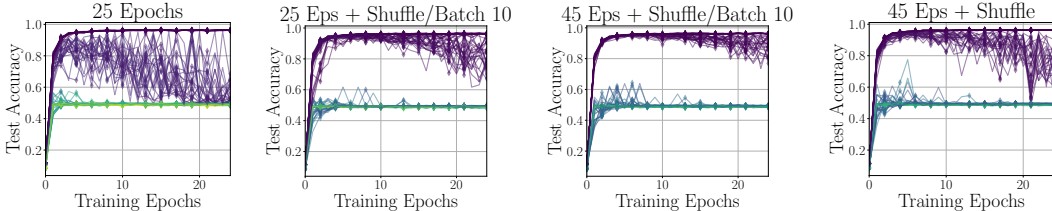

Figure 13: In the 50%-OOD setting, we ensemble for either 25 or 45 epochs. Introducing shuffling with a batch size of 10 improves performance, and increasing the number of ensemble epochs further enhances test-time accuracy. As shown in the rightmost figure, removing batching results in a performance degradation, performing similarly to the 25 Epochs + Shuffle/Batch 10 configuration. This undermines the gains achieved by extending the number of ensemble epochs.

# D    Sketching Potential Applications of Amortized Model Ensembling

---

**FedOPT (Algorithm 2) as Nested Ensembling**

**(Step 0)** Initialize the global model $x_0$, and distribute to $S$ participating clients for federated learning. Repeat the following two steps until termination, were we assume full participation purely for clarity of exposition.

**(Step 1)** After each client has trained for $K$ local epochs, view each model $x_{i,K}^t$ as an ensemble ingredient. Here, $i \in [S]$ is client index and $t$ is the global communication round timestep. Choose the pivot to be the (informative) latest global model prior to update, $x_{pivot} = x_{t-1}$. Perform a *linear* souping with all received client models $\{x_{1,K}^t, \ldots, x_{S,K}^t\}$, i.e. unamplified **GD Ensembling** with learning rate 1:

$$w_t = x_{t-1} - \frac{1}{S} \sum_{i \in [S]} (x_{t-1} - x_{i,K}^t).$$

**(Step 2)** Now, we recursively layer on another ensembling step. Once again, we choose the pivot to be $x_{pivot} = x_{t-1}$, but consider the ordered global model ingredients $x_0, w_1, \ldots, w_t$. We have saved all past pivoted server-side pseudogradients for adaptive optimization, so we shall focus on obtaining the new server pseudogradient. We further time-weight the importance of newly obtained pivoted pseudogradient updates by selecting an amplification schedule $\zeta_t = t + 1$. Then at step $t$, we have for the pivoted pseudogradient

$$\Delta_t := g_{server}^t = \frac{\zeta_t}{t+1}(w_t - x_{t-1}) = \left( \frac{1}{S} \sum_{i \in [S]} x_{i,K}^t \right) - x_{t-1},$$

which is processed by the server-side ensembling optimizer to perform an adaptive update such as **Adagrad Ensembling**. This gives $x_t$, the newest global model ensemble. Afterwards, redistribute this ensemble to participating clients for the next round. We may treat server-side ensembling as an ongoing procedure where a new ingredient $w_t$ is added at each timestep $t$, from which a new pivoted pseudogradient is calculated and amplified. To reflect the ongoing nature of this process, we call this *stewing*.

---

There are many applications of amortized model ensembling, and any setting which may benefit from aggregating or averaging DNN model parameters in the weight-space is relevant. We note that DNN training generally involves a hyperparameter sweep over an extensive parameter grid, which will necessarily output a variety of mature learners drawn from the distribution $\mathcal{D}$ of trained models which can be used as ensembling ingredients. Thus, various meta-optimization strategies can be developed to combine the residual models after the hyperparameter sweep.

As noted in Section 2.2, previous works have shown that models sharing a portion of the optimization trajectory are highly amenable to linear model weight averaging. Some approaches to induce (par-

tially) shared trajectories include taking multiple snapshots of state dictionaries along a single training run [25, 18], starting training from the same random initialization [26, 27], and/or initializing with the identical pre-trained model for fine-tuning or transfer learning [3, 22, 23]. Another technique to ensure model optimization trajectory is shared is federated learning [49], where multiple client models are aggregated in the server into a singular learner, then redistributed to all participating clients in the following round using various model update methodologies [50, 51, 52, 53, 54, 55]. In particular, each client may have data shards from diverse, heterogeneous private distributions, use different random seeds, or even alternative optimizer strategies (e.g., Federated Blended Optimization [31]). Thus, in order to provide a more specialized usage setting, we briefly discuss applications in federated learning.

**Distributed Learning.** Distributed learning often involves combining multiple trained models from clients optimizing over localized data shards, sampled from variant data distributions [56, 57, 58, 59]. The aggregation of the client models are typically performed in a central server with substantial computational resources, which is then redistributed to clients for further training. The standard aggregation step in the server is a simple average in popular algorithms such as FedAvg [49], which corresponds to souping the client models prior to redistribution. As we show in Section 4.1, utilizing more amortized model ensembling epochs during server-side model aggregation can enhance performance compared to standard souping, possibly closing the existing performance gap between federated learning and centralized training in practice [60]. We especially expect improvements when each model is being trained on isolated data shards prior to aggregation, as is the case in client-diverse distributed DNN optimization scenarios such as cross-device personalized mobile models [56]. Additional applications for the architecture and optimizer-agnostic zero-data amortized model ensembling framework may be found in large-scale DNN training [61, 62] or privacy-sensitive applications such as healthcare [63, 64, 65], where preserving the domain knowledge of rare classes is essential, and retraining on data incurs significant computational costs as well as data-leakage risks.

**Federated Learning (FL).** In more typical FL setups, while there are far more clients, the participation rate is considerably lower. Furthermore, due to local resource limitations, each transmitted client model after local training is a weak learner during early communication rounds, that has only been educated through a small number of gradient updates. After each communication round, client weights are sent to the server for aggregation, where model ensembling may be applied. In particular, the state-of-the-art FedOPT [66] framework (Algorithm 2) may be understood as a nested adapative-linear ensembling strategy, which we have verbally described above for added clarity.

Viewed from this perspective, we may modify any layer of the nested ensembling detailed in FedOPT **(Steps 1-2)** to derive a more general form of adaptive federated optimization. For example, in **(Step 1)**, we may consider an amortized ensembling strategy instead of linear souping, and in **(Step 2)** we may consider using a different amplification schedule $\zeta_t$ or an alternative pivot (e.g., ensembled pivot). This gives rise to a very diverse range of new algorithms (Algorithm 3), which to our knowledge have not been previously explored in the literature. For ease of notation, we informally refer to amortized ensembling in Algorithm 3 as 'souping', for consistency with the algorithm name FedSoup. We leave the evaluation of such derived algorithms for future work in distributed learning.

---

| **Algorithm 2** FedOPT (Simplified) | **Algorithm 3** FedSoup (Simplified) |
|---|---|
| **Require:** Initialize $x_0$, $Client/ServerOPT$ | **Require:** Initialize $x_0$, $ClientOPT$ |
| | $\qquad ClientSoup, ServerStew$ |
| 1: **for** $t = 1, \dots, T$ **do** | 1: **for** $t = 1, \dots, T$ **do** |
| 2: $\quad$ Sample subset $\mathcal{S}^t \subset [N]$ of clients | 2: $\quad$ Sample subset $\mathcal{S}^t \subset [N]$ of clients |
| 3: $\quad$ **for** each client $i \in \mathcal{S}^t$ (in parallel) **do** | 3: $\quad$ **for** each client $i \in \mathcal{S}^t$ (in parallel) **do** |
| 4: $\qquad x_{i,K}^t = ClientOPT(x_{t-1}, K$ epochs) | 4: $\qquad x_{i,K}^t = ClientOPT(x_{t-1}, K$ epochs) |
| 5: $\qquad \Delta_i^t = x_{i,K}^t - x_{t-1}$ | 5: $\quad$ **end for** |
| 6: $\quad$ **end for** | 6: $\quad w_t = ClientSoup(\{x_{i,K}^t\}_{i \in \mathcal{S}^t})$ |
| 7: $\quad \Delta_t = \frac{1}{|\mathcal{S}^t|} \sum_{i \in \mathcal{S}^t} \Delta_i^t$ | 7: $\quad x_t = ServerStew(x_0, w_1, \dots, w_t)$ |
| 8: $\quad x_t = ServerOPT(\Delta_1, \dots, \Delta_t)$ | 8: **end for** |
| 9: **end for** | |

# E  Neural Averages over CIFAR-100 Images

Images were synthesized using AME with 500 distinct random seeds, optimized via AdamW with learning rate $\eta = 5 \times 10^{-3}$ and momentum parameters $\beta_1 = 0.9$, $\beta_2 = 0.99$. The classifier ViT was fine-tuned exclusively on CIFAR-100 using AdamW with $\eta = 10^{-4}$, weight decay $\lambda = 0.05$, and the same momentum parameters. Notably, in this setting, uniformly averaged (souped) images were consistently blurry and visually uninformative (see, e.g., Figure 3), and were therefore omitted from the main visualizations.

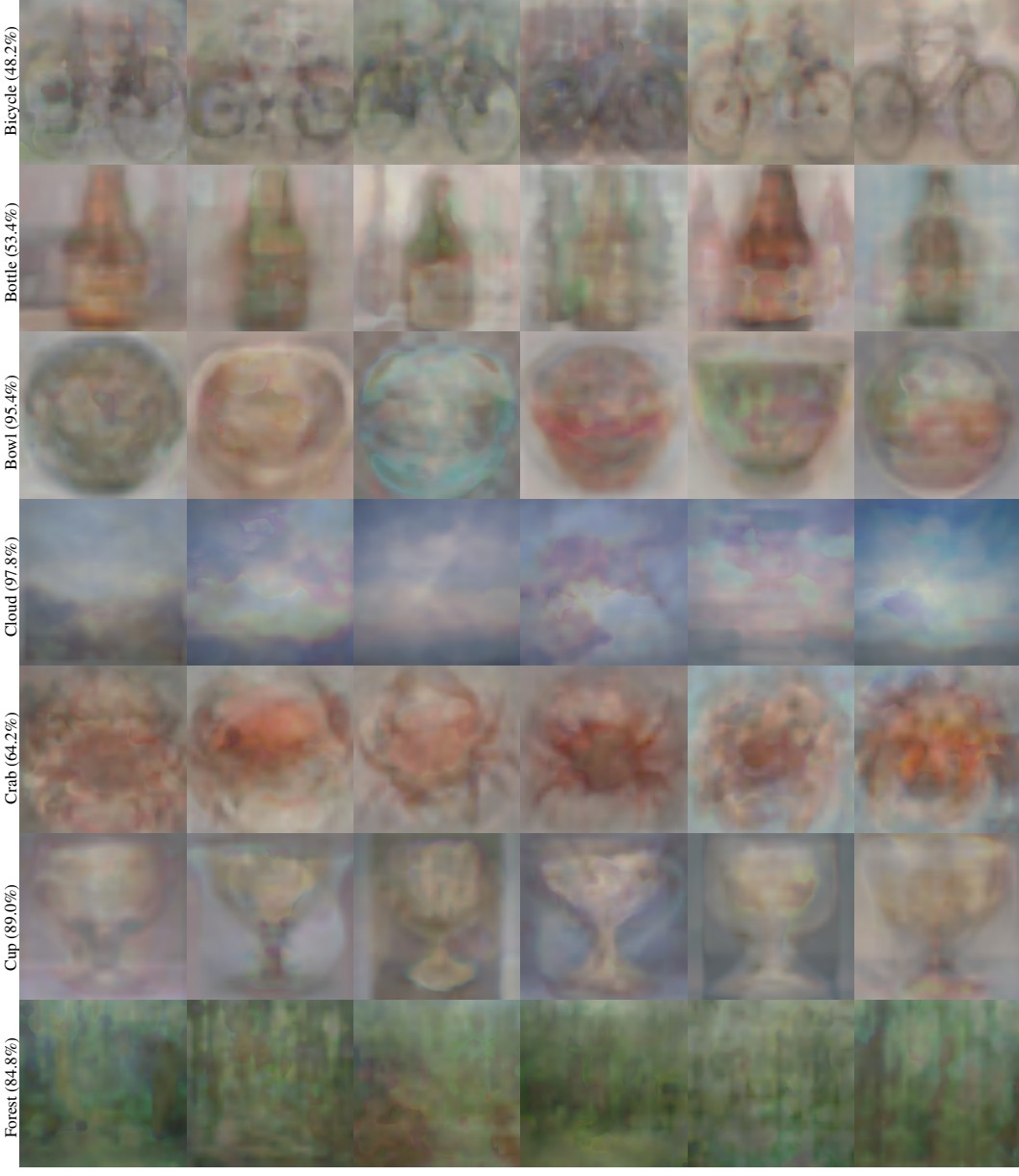

Figure 14: Neural averages of CIFAR-100 classes synthesized via AME. Percentages next to each class label indicate the proportion of AME-generated images (out of 500 per class) that were correctly classified by a ViT trained solely on CIFAR-100. This setup is essentially equivalent to ensembling a set of neural nets that give the same classification for the identity image in the weight space. For additional details, see Section 4.2.

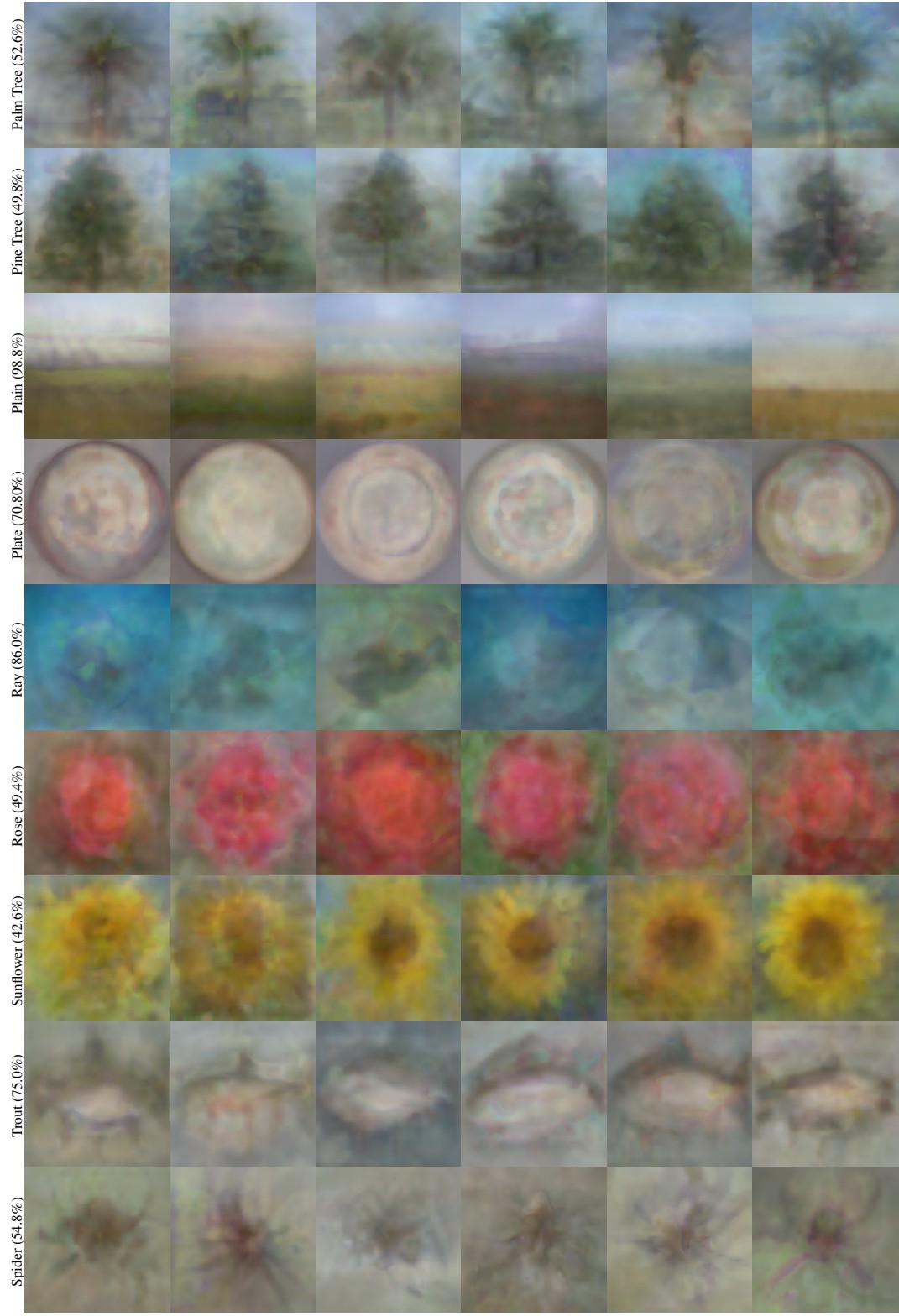

Figure 15: Neural averages of CIFAR-100 (continued). The setup is identical to Figure 14.

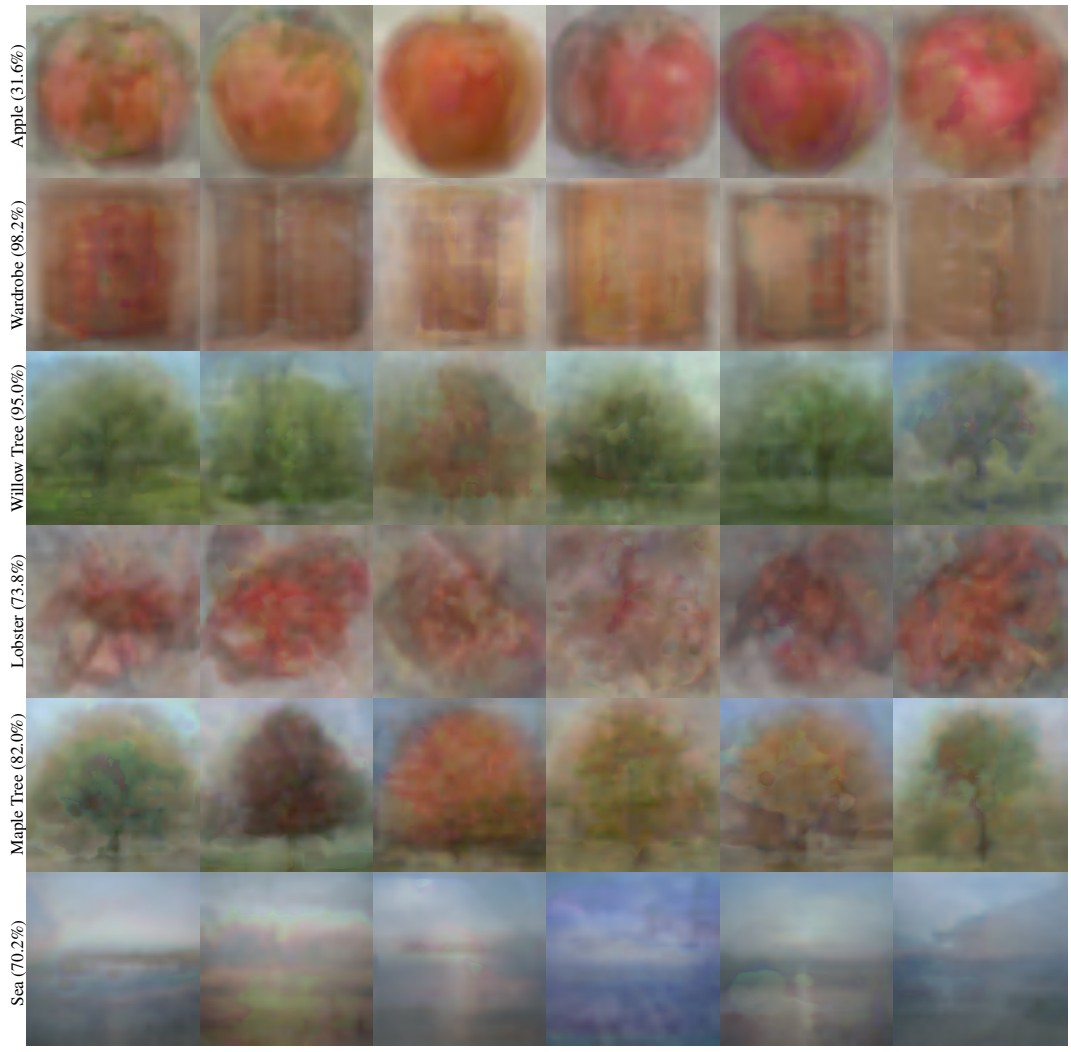

Figure 16: Additional neural averages of CIFAR-100 classes synthesized via AME. The setup mirrors that of Figure 14. While some synthesized classes (e.g., willow tree vs. maple tree) appear visually indistinguishable to humans, a standard ViT trained on CIFAR-100 classifies them correctly with high accuracy. For reference, random guessing yields a 1% success rate.

# F    Deep Learning Optimizers as Statistical Estimators via Pseudogradients

A central idea behind AME is deceptively simple: update model parameters using gradient-like signals to reduce a loss. In the context of equation (5), this might suggest that AME merely performs stochastic gradient descent on a quadratic loss, recovering the empirical average as the minimizer of (1). This naturally raises a question: *Is AME just a stochastic approximator that recovers model soup?*

The key insight in conceiving this work was that this *is not* the case. The behavior of AME is governed not by the quadratic loss alone, but by the choice of deployed optimizer as well as the distribution of materialized ensemble ingredients. Each optimizer induces specific characteristics in the optimization trajectory, often described as *implicit biases* [67, 68]. These biases cause different optimizers to converge toward distinct solutions, even when minimizing over the same loss. This phenomenon is represented in the training of deep networks across identical loss functions (e.g., adaptive optimizers like Adam often outperform SGD [11, 31, 55]) and here, we reinterpret these behaviors through a statistical and geometrical lens.

## F.1    Providing Intuition: Geometry of Adam Optimizer

To build geometric intuition, consider Adam, whose pseudocode is outlined in Section 3.2. Adam can be interpreted as computing an update direction proportional to $\mathbb{E}[\xi]/\sqrt{\mathbb{E}[\xi^2]}$, where $\xi$ represents the stochastic gradient. These expectations are approximated via exponentially decaying averages, with $\varepsilon > 0$ added to the denominator for numerical stability.

Importantly, Adam maintains moving averages over time. The numerator, an exponential average of gradients, introduces momentum and helps stabilize updates in the presence of noise. The denominator, a moving average of squared gradients, adapts the learning rate for each coordinate. If a coordinate has experienced large historical gradients, Adam reduces its step size in that dimension; if the gradients have been small, it increases it. This has a regularizing effect: in high-variance regions, Adam takes conservative steps, while allowing more aggressive updates in low-variance directions.

This behavior departs meaningfully from naive gradient descent with uniform learning rates. Rather than treating all coordinates equally, Adam imposes a form of adaptive caution based on the local geometry of the optimization landscape and the historical noise profile.

## F.2    Neural Optimizers as General Algorithms for Online Statistical Estimation

We now propose an alternative interpretation, central for conceiving this paper: *optimizers can be viewed as streaming algorithms for computing statistical estimators*. Each update involves computing a distance vector between the current estimator and a new data point (or batch), adjusted according to rules defined by the optimizer. In AME, we have interpreted these as *pseudogradients*, but this principle applies more generally for any type of problem convertible into an online inference setup. This framework naturally accommodates batching and allows us to define an entire family of estimators parameterized by the optimizer's hyperparameters.

From this perspective, AME is not simply performing naive backpropagation on a fixed quadratic, but is instead instantiating a *family* of estimators whose properties are shaped by the implicit geometry of the optimizer. In the synthetic experiments below, we validate this interpretation empirically and show that AME can outperform the empirical average, especially in heavy-tailed or noisy regimes–linking back to the discussion in Section 3.1.

## F.3    Validating Heuristics of Statistical Estimation and Optimization

There are many distributions for which the empirical average does not coincide with the maximum likelihood estimator (MLE). Notable examples include the Laplace, Cauchy, and Pareto distributions. In particular, the Cauchy distribution lacks a defined mean and variance; its MLE must be estimated numerically, and the sample median is often used as a robust proxy [69]. This distinction becomes important in the context of AME, which behaves like an optimizer and does not necessarily converge to the empirical average (i.e., model soup).

In this experiment, we draw 60,000 samples from a standard Cauchy distribution and treat this as the 'population' from which observations can be drawn. From this, we subsample 300 points to simulate the ensemble ingredients, analogous to materializing 300 two-dimensional model weights. We then compute both the model soup average and the AME-ensembled result (using Adam). The AME ensembles are deliberately initialized at the coordinates $(10, 10)$, far from the MLE estimate to create an adversarial starting point, placing AME at a disadvantage by design. This trial is repeated 300 times, and the results are visualized in Figure 17 (top). The identical experiment is repeated for the standard Gaussian distribution, a non-heavy-tailed distribution, in Figure 17 (bottom).

The visualization reveals that model soup exhibits high variance and completely fails to converge toward the proxy MLE under subsampling in heavy-tailed regimes. In contrast, Adam-AME produces estimates that diverge from the empirical average but reliably converge near the proxy MLE. By contrast, GD-AME, which necessarily must only linearly interpolate between the model ingredients, demonstrates similar properties to model soup estimators. This confirms that AME functions as an optimizer-based estimator rather than a simple average, and can yield improved performance depending on the underlying distribution. That is, an optimizer's output is itself an estimator, and under certain distributions (e.g., heavy-tailed or skewed), AME may outperform soup by aligning more closely with a robust statistical objective, especially when instantiated with an adaptive optimizer.

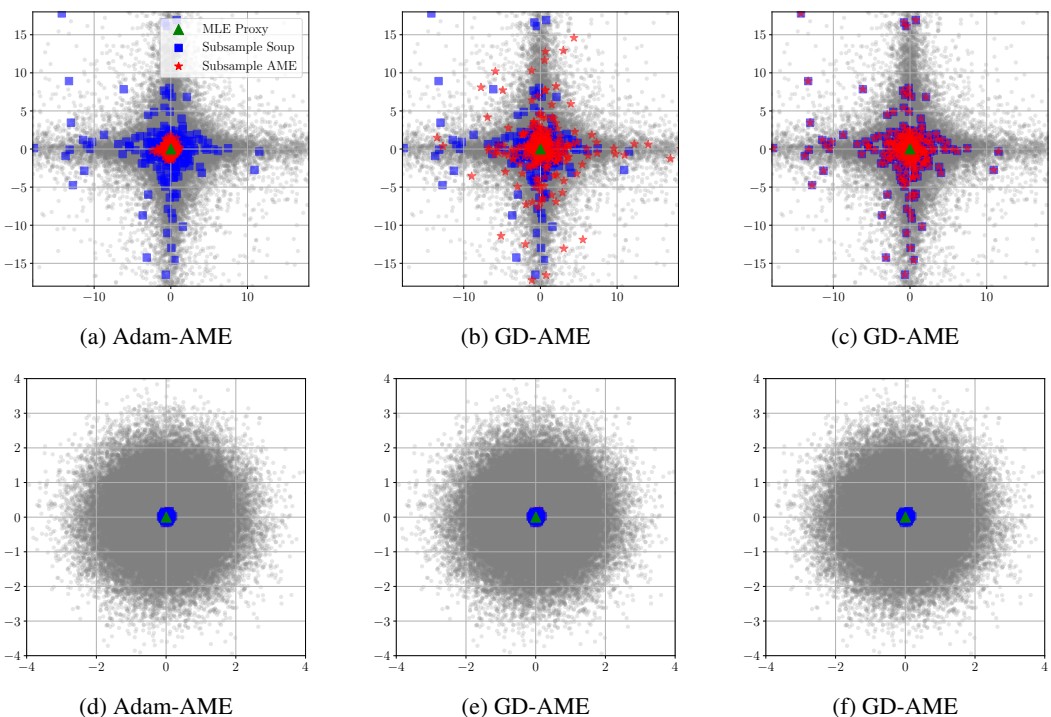

Figure 17: Each panel compares AME ensembles (Adam or GD) against model soup using identical data samples per trial. (Top) Results under the heavy-tailed Cauchy distribution. In (a), AME uses Adam with $\eta = 0.1$, $\beta_1 = \beta_2 = 0.2$, $\varepsilon = 10^{-8}$. In (b–c), AME uses gradient descent (GD) with $\eta = 1$. In (c), the GD learning rate decays as $\mathcal{O}(1/t)$, where $t$ is the number of backpropagation steps. (Bottom) Analogous experiments under the Gaussian distribution. Hyperparameters match those in (a–c) except for reduced learning rates: $\eta = 0.01$ in (d) and $\eta = 0.1$ in (e–f). All experiments use batch size 20 and 200 ensemble epochs. In (d–f), AME ensembles (red) align closely with the soups (blue) centered around the green MLE, and are therefore visually occluded.

## F.4 Intuitions from Synthetic Experiments

The central goal of AME is to harness insights from optimization theory to construct a novel class of empirical estimators that go beyond simple linear averaging. Rather than merely approximating the empirical mean, AME leverages existing optimizers to synthesize ensemble models that encode richer statistical structure–structures often missed by methods such as model soup. Empirically,

certain optimizers (e.g., Adam) outperform others (e.g., SGD) in this context, a behavior that may be attributed to their distinct implicit biases. Understanding these biases remains an open and active area of research [67, 68], particularly in connection to why adaptive methods perform better for training specific architectures such as transformers [10, 11, 32, 38, 39].

In the case of AME, our aim is not to find the global minimizer of a convex loss–an outcome that would simply reproduce model soup–but to exploit the optimizer's trajectory through weight space in order to bias the ensemble toward regions that better reflect the underlying data-generating distribution. This is empirically supported in Figure 17, where we demonstrate that both the choice of optimizer and its hyperparameters (e.g., learning rate decay) yield qualitatively different ensemble behaviors. These results suggest that AME should be viewed not as a single objective minimizer, but as defining a flexible family of estimators shaped by the inductive biases of the optimization process.

This reinterpretation positions adaptive optimizers as numerical routines for computing robust, noise-aware estimators. By initializing at a fixed point and applying pseudogradient updates toward sampled models or datapoints, AME produces solutions that converge closer to the likelihood maximizer than the empirical average. We believe this perspective motivates the development of future deep neural network optimization-inspired ensembling techniques as statistical estimators in their own right.

