# OpenReview forum: "On Defining Neural Averaging"
_NeurIPS.cc/2025/Workshop/UniReps — UniReps2025 oral_

### Official Review · Reviewer_RVEd · 2025-09-04

**Confidence:** 3

**Review:**

## Summary
This work explores neural network weight averaging through the lens of (pseudo) gradient descent. This novel perspective opens to door to adaptive optimizers acting as weight averaging methods. The paper explores this connection and provides experiments for merging expert image classification models and offers a way to average images directly. The key advantage of this method is the lack of data required to merge these models, setting the problem apart from routing and standard learnable model soups.

## Strengths
 - The gradient perspective is novel and opens the door to using lots of existing techniques for neural weight averaging
 - The experiments demonstrate strong performance vs. data-free baselines
 - Clear improvement from using adaptive optimizer vs standard GD

## Weaknesses
 - The evaluation is limited to image classifiers. It seems intuitive that the method would work with other classifiers, but it is unclear if this weighting method would work for generative methods, e.g. diffusion models
 - While the performance is impressive vs. data-free baselines, it would be helpful to see the "ceiling" of data-dependent model soup methods like MEHL-Soup
 - Hyperparameter optimization (early stopping, etc.) is not clear without some validation data, how can this be done?
 - The experts combined in the experiments are of equal strength, how does the method perform with some weak soup ingredients and some strong?

## Questions
 - Is there a relationship between the Gaussian interpretation of model averaging mentioned in the paper and the gradient descent interpretation? It seems there is always a connection between square loss and Gaussians!
 - Is it obvious that this should work for generative models? It seems that classifiers may combine well while generative models may produce some low quality off-manifold images

**Score:**

4

**Topic Fit:**

2

---

### Official Review · Reviewer_EysK · 2025-09-08
**Review of submission 25**

**Confidence:** 5

**Review:**

The paper analyzes optimal strategies for "model soups" (model weight averaging), by formulating the task as the minimization of a stochastic loss, and using the difference among weights as "pseudogradients" for solving this loss. The paper introduces several innovations, such as adaptive optimization algorithms in place of standard gradient descent and how to order the tasks to improve the performance.

With respect to the workshop, the topic is of interest to the community, although the paper focuses more heavily on performance and less on the similarity or impact on the neural representations obtained by the different algorithms.

My main concern is that the paper focuses only on model soups (anisotropic weight averaging), while the literature on model merging in vast and also includes techniques very similar to those discussed here - most notably, task vectors (task arithmetic) are precisely what the authors call "pivoted pseudogradients". While the results are still interesting, lack of comparisons (both in terms of related works and as baselines) with this extended literature is a major issue, and I lean towards rejection to allow the authors to incorporate them.

As a final side-note, the authors may be interested in this concurrent work, which also analyzes the relations between multitask losses, gradients, and task vectors: https://www.arxiv.org/pdf/2508.16082.

**Score:**

2

**Topic Fit:**

2

---

### Official Review · Reviewer_s4CZ · 2025-09-15
**Significant breakthrough with a sound theoretical basis requiring minor refinements**

**Confidence:** 4

**Review:**

Summary:

The paper reframes model soup as stochastic gradient descent on pretrained parameters, introducing Amortized Model Ensembling (AME) as a principled framework for neural averaging. This interpretation connects model merging and aggregation with the broader literature on adaptive gradient algorithms, enabling more flexible and effective aggregation strategies. The authors train vision transformers using grid-searched hyperparameters on the Google Landmarks dataset, demonstrating that AME consistently outperforms standard model soup while maintaining strong in-distribution accuracy. Additionally, they reinterpret CIFAR-100 images as first-layer weights, where AME achieves significantly above-chance classification on image tensors constructed from pixels sampled from a uniform distribution. They also find that the greedy strategy of immediately rejecting an ensemble due to not performing better on a held-out validation set is worse than standard AME. Lastly, the paper discusses the advantages of AME in utilizing models from hyperparameter search, maximizing the benefits of distributed learning, and approximating statistical estimators.

Strengths:

This paper makes significant progress over model soup on model merging using sound theoretical justifications that subsume previous work.

The authors consider major applications like maximizing performance gains from a hyperparameter sweep and distributed or federated learning.

The authors demonstrate that the result is a property of adaptive optimizers over the optimization landscape by testing on known distributions.

Weaknesses:

There are a few unclear expressions in the appendix possibly caused by typos:

In Proposition 1 of Section (A.1), I think $\varepsilon_i >> \sum_{j=1}^{i} g_j^2$ is required for the estimate to hold.

Within the main equation used in proving (ii) of Section (A.3), I think it should be $\sum_{t=n}^{\infty} \eta_t \frac{\zeta diam(\chi)}{N}$.

It is not clear to me, while checking section (A.3), why for $C_n$ they have $(1-\beta_1)\sum_{r=1}^{t-1} \beta_1^{t-r}g_r + (1-\beta_1)g_t$ instead of the equivalent $(1-\beta_1)\sum_{r=1}^{t} \beta_1^{t-r}g_r$. Also, do we not get $\frac{\beta_1^2}{\beta_2-\beta_1^2}$ when doing the infinite sum?

The interpretability of the figures is undermined by excessive data points instead of using summary statistics.

Ironically, the section on neural averages is the weakest part of the paper. The experimental setup resembles injecting noise or random perturbations, and it is not clear what unique insight is gained by reinterpreting images as layers.

Comments:

Due to the novelty of the work and the significant number of experiments and applications discussed, if possible, I would like to nominate this paper as a candidate for best paper. While there is room for improvement, the paper brings valuable insight that outweighs such concerns. I apologize to the area chair for submitting the review at the last minute. I wanted to spend as much time as possible deliberating on the paper and refining my review because of its importance.

**Score:**

5

**Topic Fit:**

3

---

### Official Review · Reviewer_uDpv · 2025-09-16
**AME: Generalized, data-free, Model Soups**

**Confidence:** 5

**Review:**

## Evaluation

The paper is mathematically careful and provides a clear optimization-based reinterpretation of weight averaging, making its conceptual contribution strong. The clarity is generally good, though some dense derivations and toy experiments may confuse the reader. Its originality lies in formalizing model soup as optimization and extending it with adaptive ensemble dynamics, though the idea that weight differences encode useful signals is not entirely new. The significance is promising, especially for data-free model merging and federated contexts, but empirical scope is limited to ViTs on small/medium datasets, with no comparisons to other modern merging methods, leaving its practical impact somewhat constrained. Moreover, from the reviewer’s perspective, the usage of pseudo-gradients can be connected to a certain kind of “client-update-less” Federated Learning algorithm.

## Pros

1. Connects _model soups_ with linear averaging, optimizer-based updates, and ensemble meta-optimization under a single framework.
2. AME is data-free: no training data for merging.
3. Experiments show AME outperforming both individual models and soups, especially in OoD settings.
4. AME has a very good potential to be beneficial on distributed or federated learning settings.

## Cons

1. Not clear how the method could scale well for billion-parameter models, especially for many ensembling steps $T$.
2. Results might be biased towards pivot choice and expert ordering
3. There seems nothing preventing pseudo-gradients drift toward certain experts, i.e., a potential collapse to a subset of experts. (Maybe not a full concern, but might not be a property we want to see within model merging)
4. If meta-training minima lie very far-apart, pseudo-gradients might be unstable.
5. How do pseudo-gradients compare with other ways of measuring model differences, such as task arithmetics?

**Score:**

4

**Topic Fit:**

3